

# Secondary Organic Aerosol Yields from the Oxidation of Benzyl Alcohol

Sophia M. Charan, Reina S. Buenconsejo, and John H. Seinfeld

California Institute of Technology, Pasadena, California 91125, United States

**Correspondence:** seinfeld@caltech.edu

**Abstract.** Recent inventory-based analysis suggests that emissions of volatile chemical products in urban areas are now competitive with those from the transportation sector. Understanding the potential for secondary organic aerosol formation from these volatile chemical products is, therefore, critical to predicting levels of aerosol and for formulating policy to reduce aerosol exposure. It is clear that a plethora of oxygenated compounds are either emitted directly into the atmosphere or emitted indoors

5   and later escape into the outdoors. Experimental and computationally simulated environmental chamber data provide an understanding of aerosol yield and chemistry under relevant urban conditions (5–200 ppb NO and 291–312 K) and give insight into the effect of volatile chemical products on the production of secondary organic aerosol. Benzyl alcohol, one of these volatile chemical products, is found to have a large secondary organic aerosol formation potential. At NO concentrations of ~80 ppb and 291 K, secondary organic aerosol mass yields for benzyl alcohol can reach 1.

## 1 Introduction

A major component of ambient fine particulate matter is secondary organic aerosol (SOA), the precursors of which are originally emitted into the atmosphere in the gas-phase (Shrivastava et al., 2017; Goldstein and Galbally, 2007). Through single or multiple generations of oxidation, emitted vapors can become progressively less volatile and eventually condense into the

15   particle phase to form this SOA (Seinfeld and Pandis, 2016).

Understanding the formation of particulate matter is of critical importance. Exposure to particulate matter causes respiratory and cardiovascular disease (Mannucci et al., 2015), and yet particulate matter has remained stubbornly high despite regulation: over 20 million people in the U.S. live in regions with larger concentrations of $PM_{2.5}$ than deemed safe (EPA, 2012). Additionally, SOA-containing particles can serve as cloud condensation nuclei; the interaction between particulate matter and cloud

20   formation is one of the most important processes in the Earth's radiative budget and, therefore, in climate predictions (IPCC, 2014).

However, accurately predicting the mass of secondary organic aerosol formed from the oxidation of volatile chemical products (VCPs) poses a major challenge. A mass-balance analysis of VCPs in the Los Angeles atmosphere indicates that VCPs





could account for around half of the SOA in that area (McDonald et al., 2018). This analysis was based on estimating sec-
ondary organic aerosol yields for a number of these oxygenated compounds that have traditionally not been studied for their
SOA formation potential. Direct measurements of the SOA yields of these compounds is paramount to constraining estimates
and formulating policy to reduce secondary organic aerosol formation (Burkholder et al., 2017).

This study focuses on one of these volatile chemical products, benzyl alcohol. Benzyl alcohol is a widely used compound in
consumer products that can be found in soaps, inks, paints and, correspondingly, indoor air (Wang, 2015; Harrison and Wells,
2009). It is also emitted from biogenic sources, such as fruits and flowers (Baghi et al., 2012; Bernard et al., 2013; Horvat
et al., 1990). The emission inventory-based analysis by McDonald et al. (2018) of the production rates of volatile chemical
products estimated that benzyl alcohol comprised 0.06% of the total volatile organic compounds (VOCs) in the Los Angeles
basin in 2010. Using the Statistical Oxidation Model, they calculated that for half a day of oxidation under high ambient $NO_x$
conditions, benzyl alcohol will have a SOA yield of 0.09. Based on this value, it was further estimated that benzyl alcohol
contributes 0.14% of the total atmospheric secondary organic aerosol in the Los Angeles basin.

Whereas the SOA yield of benzyl alcohol oxidation estimated in the McDonald et al. (2018) analysis was relatively low, in
a laboratory chamber study, Carter et al. (2005) measured the SOA yield of benzyl alcohol to be ~0.3 in a mixture of reactive
compounds and 25–30 ppb of $NO_x$. This reactive compound mixture comprised compounds that one would not expect to form
significant SOA yield, but that may influence the fate of $RO_2$ radicals that could be formed from benzyl alcohol oxidation. That
study also estimated the reaction rate constant of benzyl alcohol with OH as $2.56 \times 10^{-11}$ cm$^3$ molec$^{-1}$ s$^{-1}$. An extension of
the study (Li et al., 2018), which also used a base mixture of reactive compounds, determined a benzyl alcohol SOA yield of
0.41.

The goal of determining SOA formation in an environmental chamber is to extrapolate the results to the atmosphere. Since at
different times or in disparate places, different temperatures or $NO_x$ mixing ratios may be most relevant, it is important to study
SOA formation in a wide parameter-space. Studies performed under varying conditions can also assist in teasing out which
data result from the atmospheric chamber itself and how these data ought to be corrected for the atmosphere. For example, for
toluene, a compound for which benzyl alcohol is a major photooxidation product (Hamilton et al., 2005), Zhang et al. (2014)
found a SOA yield 70% higher at low $NO_x$ concentrations than at high $NO_x$ concentrations and found that the true SOA yield
was a factor of 4 higher than that calculated without accounting for the chamber-process of vapor wall deposition.

## 2 Instruments and procedure

### 2.1 Experimental method and chamber description

All experiments were performed in batch mode in the Caltech 17.9 m$^3$ FEP Teflon-walled Environmental Chamber. The
chamber volume was characterized according to the procedure outlined in Schwantes et al. (2017a). While the chamber pressure
remains constant throughout the duration of an experiment, the volume decreases as air is sampled by various instruments; the
fraction of the volume at the end of the experiment compared to the beginning of it is given in Table 1. Before each experiment,
the chamber was flushed for > 24 h with clean air (compressed air with ozone, nitrogen oxides, water vapor, and organic





carbon removed). The radical source $H_2O_2$ was injected at 42°C and 5 Lpm into the chamber, followed by the injection of benzyl alcohol (Sigma Aldrich ReagentPlus, ≥99%) with gentle heating (60°C) at 2 Lpm (5 Lpm for experiments S1–3 and E1) for >50 min. The purity of the benzyl alcohol was verified with Nuclear Magnetic Resonance (NMR) spectroscopy.

Meanwhile, a 0.06 M $(NH_4)_2SO_4$ solution (0.15 M for experiments S2 and E1) was atomized and the resulting particles dried, charge-conditioned with a TSI Model 3088 soft x-ray neutralizer, and then injected into the chamber for varying lengths of time (depending on the desired initial seed concentration; note that no particles were injected for experiment S1). The solution was sonicated before each injection. Then, NO (506.9 ppm ± 2%, Airgas Specialty Gases, Certified Standard) or, for experiment E1, $NO_2$ (488 ppm, Air Liquide) was injected into the chamber at 5 Lpm to achieve the desired initial NO or $NO_2$

concentration. Ultraviolet broadband lights centered around 350 nm were used to photolyze $H_2O_2$. The $NO_2$ photolysis rate, $j_{NO_2} = 6.2(\pm0.1) \times 10^{-3}$ s$^{-1}$, was measured using a 0.29 L quartz tube and the procedure outlined in Zafonte et al. (1977).

A Vaisala HMM211 probe was used to measure the temperature and humidity of the chamber. Humidity was calibrated for RH from 11 to 95% (using LiCl, $KNO_3$, $Mg(NO_3)_2$, and $MgCl_2$ salts). A Teledyne Nitrogen Oxide Analyzer (Model T200) was used to measure the NO and $NO_2$ concentrations throughout the experiments; note that this instrument measures the

contribution of $NO_y$ compounds (e.g., organic nitrates) as $NO_2$. Owing to some drift between experiments, linear fits were performed on the slope and offset calibrations, except for experiments S2–3 and U5, due to a calibration problem. Ozone was measured with a Horiba Ambient Monitor. NO, $NO_2$, and $O_3$ measurements were recorded every 30 s. Humidity and temperature uncertainties were calculated as standard deviations from the mean value, where measurements were taken every 30 s throughout the experiment. Initial NO and $NO_2$ mixing ratios were determined (as well as their standard deviations) prior

to irradiation during the background collection period (usually ≳60 min). For experiments N1–6 and U6, NO was continuously injected during oxidation to maintain a stable NO mixing ratio.

## 2.2   Gas-phase measurements

A $CF_3O^-$ chemical ionization mass spectrometer (CIMS), operated in the negative mode, measured oxidation products and the benzyl alcohol concentration by scanning m/z ratios between 50 and 330. The CIMS is equipped with a Varian 1200 triple

quadrupole mass analyzer. A custom-built inlet was used to ensure that the sample was taken at a constant temperature (the top of the inlet was 25°C). To reduce loss of vapor to the tubing prior to analysis, the CIMS sampled off of a bypass flow that was accelerated using a mechanical pump.

The 193 m/z signal (the mass of benzyl alcohol + $CF_3O^-$), which was measured every 162 to 172 s, was normalized to the 86 m/z signal (the M+1 peak for $CF_3O^-$) and used to measure the benzyl alcohol concentration. This signal was calibrated using

dilutions of an 800 L Teflon bag of ~44 ppb benzyl alcohol. The concentration in this bag was verified using Fourier transform infrared absorption (FT-IR) spectroscopy with a 19 cm path length and absorption cross sections from the Pacific Northwest National Laboratory (PNNL) database. In this way, any wall or sampling loss was accounted for since the CIMS sampled from the same volume as the FT-IR. Multiple FT-IR samples were taken until each spectrum gave the same concentration; this was to ensure a minimal effect from any compound deposited on the instrument walls.





**Table 1.** Experiments analyzed

| Label/Day | [BnOH]$_0$ (ppb) | T (K) | Initial Seed Surface Area‡ ($10^3$ μm$^2$ cm$^{-3}$) | [NO]† (ppb) | Wall-Loss Slope (μm$^3$ cm$^{-3}$ s$^{-1}$) | $k_{\text{BnOH+OH}}$[OH] ($10^{-4}$ s$^{-1}$) | Length (h) (% of Total Volume at Experiment End) | SOA Y (ω = 0) | SOA Y (ω = 1) |
|---|---|---|---|---|---|---|---|---|---|
| R1/190321 | 199±32 | 291.0±0.3 | 1.74±0.17 | 77.3±0.9 | 0.048±0.050 | 1.10 ± 0.06 | 6.1 (85.9%) | 0.76±0.16 | 0.79±0.16 |
| R2/190323 | 160±18 | 290.9±0.3 | 1.98±0.18 | 77.4±0.8 | -0.041±0.145 | 1.03 ± 0.06 | 6.5 (85.0%) | 0.99±0.16 | 1.04±0.16 |
| R3/190312 | 202±24 | 291.1±0.2 | 1.50±0.16 | 72.6±0.7 | -0.027±0.042 | 0.86 ± 0.04 | 12.0 (72.7%) | 0.70±0.13 | 0.75±0.13 |
| R4/190319 | 199±28 | 291.0±0.2 | 1.97±0.18 | 74.0±1.0 | -0.009±0.076 | 1.03 ± 0.06 | 6.3 (85.3%) | 0.79±0.15 | 0.83±0.15 |
| R5/190128 | 222±27 | 291.2±0.2 | 2.19±0.21 | 93.7±0.7 | -0.017±0.059 | 0.71 ± 0.03 | 8.8 (79.7%) | 0.72±0.13 | 0.78±0.13 |
| S1/191219 | 455±29 | 291.3±0.2 | 0.00±0.00 | 72.4±0.6 | | 0.49 ± 0.03 | 5.3 (89.9%) | 0.45±0.06 | 0.47±0.06 |
| S2/191002 | 252±16 | 291.2±0.2 | 0.33±0.07 | ~96 | -0.008±0.013 | 0.99 ± 0.04 | 6.3 (87.5%) | 0.39±0.04 | 0.41±0.04 |
| S3/190930 | 174±15 | 291.0±0.2 | 0.64±0.10 | ~90 | 0.016±0.017 | 1.17 ± 0.05 | 4.5 (91.0%) | 0.52±0.06 | 0.54±0.06 |
| S4/190325 | 153±27 | 291.0±0.3 | 5.47±0.32 | 77.8±0.8 | 0.010±0.213 | 1.08 ± 0.09 | 5.1 (87.8%) | 0.96±0.25 | 1.04±0.25 |
| T1/190419 | 216±30 | 296.7±0.4 | 2.33±0.21 | 75.6±0.9 | -0.069±0.062 | 1.44 ± 0.07 | 5.0 (91.1%) | 0.60±0.11 | 0.63±0.11 |
| T2/190417 | 193±23 | 301.6±0.4 | 1.93±0.19 | 71.7±0.9 | -0.012±0.060 | 1.44 ± 0.08 | 5.0 (90.9%) | 0.54±0.09 | 0.57±0.09 |
| T3/190422 | 212±34 | 306.6±0.4 | 2.76±0.23 | 76.9±0.7 | 0.070±0.144 | 1.13 ± 0.09 | 6.3 (88.9%) | 0.63±0.13 | 0.67±0.13 |
| T4/190410 | 266±43 | 311.6±0.5 | 2.12±0.2 | 80.4±0.8 | -0.013±0.114 | 1.18 ± 0.08 | 5.5 (90.3%) | 0.37±0.08 | 0.39±0.08 |
| N1/190408* | 191±27 | 291.1±0.3 | 2.00±0.19 | 4.8 (0.7–8) | 0.056±0.101 | 1.27 ± 0.05 | 5.0 (90.9%) | 0.70±0.12 | 0.73±0.12 |
| N2/190403* | 190±35 | 290.9±0.3 | 2.09±0.19 | 14.3 (8–18) | 0.003±0.094 | 1.02 ± 0.11 | 5.0 (88.1%) | 0.68±0.16 | 0.71±0.16 |
| N3/190426* | 166±32 | 290.9±0.3 | 2.71±0.23 | 64.0 (56–69) | 0.027±0.070 | 0.77 ± 0.06 | 6.0 (89.6%) | 0.66±0.17 | 0.70±0.17 |
| N4/190401* | 183±17 | 291.0±0.3 | 1.84±0.18 | 76.2 (52–106) | 0.008±0.059 | 0.86±0.05 | 5.0 (88.2%) | 0.60±0.09 | 0.63±0.09 |
| N5/190424* | 167±19 | 290.9±0.3 | 2.84±0.23 | 111.7 (103–118) | 0.027±0.186 | 0.77 ± 0.05 | 5.0 (91.1%) | 0.54±0.10 | 0.58±0.10 |
| N6/190405* | 189±18 | 290.9±0.1 | 1.78±0.18 | 200.6 (194–208) | 0.000±0.082 | 0.76 ± 0.03 | 5.0 (88.0%) | 0.47±0.08 | 0.50±0.08 |
| E1/200109^ | 295±18 | 291.1±0.2 | 2.83±0.22 | 1.4±1.0 | 0.091±0.093 | 0.83 ± 0.02 | 5.5 (89.5%) | 0.35±0.05 | 0.38±0.05 |
| L1/190110^ | 135±12 | 285.78±0.03 | 2.58±0.21 | 80.4±1.1 | 0.033±0.009 | 0.115 ± 0.002 | 16.7 (58.4%) | 0.37±0.18 | 0.51±0.18 |
| U1/190327 | 189±22 | 290.9±0.2 | ~4.03 | 81.1±0.7 | | 2.09±0.25 | 5.2 (88.0%) | | |
| U2/190430 | 136±20 | 291.1±0.2 | 1.36±0.13 | 71.0±0.9 | | 1.16±0.07 | 5.2 (90.6%) | | |
| U3/190628 | | 291.2±0.4 | ~1.48 | 77.7±0.9 | | | 5.0 (90.6%) | | |
| U4/190529 | 139±26 | 291.1±0.3 | ~5.40 | 70.7±0.7 | | 1.10±0.06 | 5.5 (89.9%) | | |
| U5/190828 | 325±20 | 284.5±0.1 | 1.70±0.14 | ~69 | | 0.19±0.01 | 5.4 (86.5%) | | |
| U6/190428* | 152±25 | 291.1±0.2 | 3.11±0.23 | 137.8 (133–144) | | 0.74± 0.06 | 5.9 (89.7%) | | |
| U7/190225 | | 290.9±0.2 | ~2.2 | 71.6±1.0 | | | 6.6 (84.1%) | | |
| U8/190227 | | 290.9±0.3 | | 76.9±0.9 | | | 9.6 (77.8%) | | |

*For these experiments, [NO] was held constant through a continuous injection.

†For constant [NO] experiments, the average [NO] is reported along with the range of [NO] throughout the experiment. For all other experiments, the initial [NO] is given with the standard deviation during the background collection period. For experiments with NO$_x$ measurement problems, an approximate value is given.

‡Experiments with particles outside the range of the SMPS used for particle measurement or those with other measurement issues are reported without error bars and should be taken as approximate values.

^Experiment E1 had an initial NO$_2$ mixing ratio of 71.0±0.8.





During the background collection period of ~1 h for each experiment, the standard deviation of the benzyl alcohol mixing ratio, along with the uncertainty in the calibration, was used to estimate the uncertainty of the initial benzyl alcohol mixing ratio (see Table 1). This combined standard deviation was also considered as the uncertainty in the measurement of the time-resolved gas-phase mixing ratio throughout the experiment. The SOA yield is determined from the reacted benzyl alcohol, which is the difference between the measured benzyl alcohol concentration at any given time and the initial benzyl alcohol concentration.

The variance of the reacted benzyl alcohol is the sum of the variances of the initial and measured benzyl alcohol mixing ratios. The uncertainty reported in Table 1 is, then, the square root of the reacted benzyl alcohol mixing ratio variance.

    The conversion from mixing ratio to mass concentration of reacted benzyl alcohol was performed assuming a constant pressure of 1 atm. Note that the chamber is located three floors from a weather station, which reported an average atmospheric pressure of 0.97 atm in the year 2019 (TCCON Weather Data, 2020); thus, 1 atm is a reasonable estimate of the pressure in the

experiments.

### 2.3   Particle-phase measurements

To measure the particle size distribution, a custom-built scanning mobility particle sizer (SMPS) with a 308100 TSI Differential Mobility Analyzer (DMA) and a TSI 3010 *t*-butyl alcohol condensation particle counter (CPC) was used with a sheath flow rate of 2.64 Lpm, an aerosol flow rate from the chamber of 0.515 Lpm, and a dilution flow of 0.485 Lpm. A full size-scan was

collected every 5.5 minutes (for experiments S1–3 and E1 scans were performed every 6 min), and the voltage was scanned over 4 min from 15 to 9875 V. Data inversion was performed using the method described in Mai et al. (2018). Total number, volume, and surface area concentrations were determined assuming 431 size bins between 22 and 847 nm. When the sample flow was <0.515 Lpm, an adjustment to the total number concentration was performed to account for the sampled flow. Particles were charged with a 500 microcurie $Po^{210}$ source, except for experiments S1–3 and E1, which used an X-ray source.

When the aerosol size distribution was close to the edges of the measurable range, a logarithmic fit of the distribution tail was performed on the edges of the distribution: diameters of 382 to 600 nm were used to fit particles above 600 nm, and those with diameters 35 to 200 nm were used to fit particles with diameters smaller than 35 nm. Fits of the tail distribution were performed on the upper end of the size distribution for experiment N5, which produced an average of a 3.4% decrease from the raw measurement in the volume concentration; the lower end of the size distribution for experiment S2, which led to a volume

concentration adjustment of <0.1%; and on both the upper and lower ends of the size distribution for experiment S1 (the nucleation experiment), which (for those points after at least 100 min of oxidation) led to a volume concentration difference of <1% from that measured in the absence of any adjustment. Particle volume was converted to particle mass with a SOA density of 1.4 g cm$^{-3}$, consistent with past work on isoprene (Dommen et al., 2006; Kroll et al., 2005, 2006) and on benzyl alcohol (Li et al., 2018).

Uncertainty in the particle size was assumed not to exceed 2 nm, based on a comparison with the SMPS at the University of California, Riverside. For the CPC-associated margin of error, according to approximate Poisson statistics, the uncertainty of the number in each particle size bin was taken as the square root of the number concentration in that bin and that value of uncertainty was propagated into surface area and volume measurements both by bin and, eventually, for the total number





concentration. Additionally, an uncertainty in the measured volume concentration due to sample noise was added from the

uncertainty of the wall-loss corrected volume concentrations in the background collection period prior to lights on (see Sect. 3.2.1).

Aerosol-phase bulk composition was determined using an in situ high-resolution time-of-flight aerosol mass spectrometer (AMS, Aerodyne Research) in the high-sensitivity V-mode. Data were analyzed with Igor Pro (version 6.37) and the Squirrel (1.57l) and Pika (1.16l) toolkits. Elemental composition was determined following the improved-ambient method from Cana-

garatna et al. (2015) and Aiken et al. (2008). Absolute uncertainties of O:C and H:C ratios are ±28% and ±13%, respectively (Canagaratna et al., 2015).

Measurements from the AMS can be utilized to determine the mass fraction of organonitrates ($RONO_2$) in the aerosol-phase following the method described by Farmer et al. (2010). Both inorganic and organic nitrates fragment to an m/z of 30 ($NO^+$) and an m/z of 46 ($NO_2^+$), but the ratio of these two fragments for organonitrates (including those derived from aromatic

hydrocarbons) and for ammonium nitrate is quite different and this difference can be utilized to determine the contribution of organonitrates to the nitrate signal in the AMS (Farmer et al., 2010; Fry et al., 2013; Kiendler-Scharr et al., 2016; Sato et al., 2010). Note that fragments of the form $C_xH_yN_z^+$ are sufficiently scarce that they are neglected (the N:C ratio was never more than 0.026 for the experiments considered here).

The measured mass ratio of $NO/NO_2$ (called the $NO_x^+$ ratio) is calibrated for ammonium nitrate for experiments R4 and

U7–8 (3.20±0.04) and is assumed for organonitrates (7.2±1.1). The organonitrates ratio was calculated using the ammonium nitrate ratio and the correlation derived by Fry et al. (2013). From this $NO_x^+$ ratio, the time-resolved ratio of the fraction of the nitrate signal that comes from organonitrates for each experiment ($x_{ON}$) can be obtained using Eq. 1 in Farmer et al. (2010). With the mass concentration of nitrates ($m_{NO_3}$) and the mass concentration determined to be organics ($m_{Org}$), the time-resolved organonitrate mass fraction of the aerosol is $\frac{x_{ON}*m_{NO_3}}{x_{ON}*m_{NO_3}+m_{Org}}$. This is plotted in Sect. 4 and in Fig. A1.

For experiments N1–3 and U1–6, the chemical composition of particle-phase compounds was further analyzed using offline ultra-high performance liquid chromatography electrospray ionization quadruple time of flight mass spectrometry (UPLC/ESI-Q-ToFMS) (Zhang et al., 2016). Post-oxidation samples were taken using 47 mm Pall Teflon filters, which were collected for ≥2 hours at 6.5 Lpm using an upstream activated carbon denuder. Additional Teflon filters were collected during photooxidation at 2 Lpm. This experimental set up is described by Nguyen et al. (2014).

The SOA collected was extracted by placing each filter sample into 6 mL of milliQ water and agitating the samples on an orbital shaker for 1 h. In an effort to prevent on-filter chemistry from occurring, samples were stored at -14°C after initial collection and before extraction. Analysis using UPLC-MS was carried out in negative mode (where the parent molecule is observed at M-H) which is sensitive to the nitroaromatics formed in the aerosol-phase. The 12 min eluent program for UPLC-MS and MS/MS fragmentation analysis required 4 μL of sample with gradient eluents between a 0.1% formic acid/99.9% water

solution and a 100% acetonitrile solution. The total flow rate was 0.3 mLpm, and masses were scanned from m/z = 7 to 4000. MassLynx software was used to analyze the resulting spectra, which calculates possible chemical formulas based on masses quantified during analysis. Mass assignments were limited to carbon-, oxygen-, and nitrogen-containing formulas as these were the only chemically viable formulas for benzyl alcohol oxidation chemistry. The structures assigned to chemical formulas





from MassLynx analysis were based on structures that corresponded to expected oxidation products and were confirmed based
on MS/MS fragmentation analysis. Isomeric analysis was not conducted for these compounds, thus structures in Table A1
represent just one possible isomer. Several experiments with similar reaction conditions (U1–4; see Table 1) were analyzed to
probe reproducibility of this technique; these experiments showed consistent results.

Other organic compounds may be present in the SOA collected that is insoluble in the extractant solvent, not able to elute
from the chromatographic column, or not detectable in negative ion mode (Surratt et al., 2008). Additionally, the UPLC-MS
exhibits different sensitivities to compounds depending on the polarizability of the compound as well as its ability to ionize. It
is likely that the UPLC-MS is quite sensitive to the nitroaromatics reported in this work as compared to other compounds.

## 3 Calculations of SOA yield

### 3.1 Method

The secondary organic aerosol yield (SOA Y) is given by

$$Y = \frac{\Delta \text{SOA}_{\text{meas}}}{\Delta \text{BnOH}_{\text{meas}}} \tag{1}$$

where $\Delta \text{SOA}_{\text{meas}}$ is the difference between the measured and wall-deposition-corrected aerosol mass concentration at a given
time and the aerosol concentration prior to the beginning of oxidation. $\Delta \text{BnOH}_{\text{meas}}$ is the reacted mass of benzyl alcohol; that
is, the difference between the initial concentration and the measured concentration at a given time.

This SOA yield calculation uses $\Delta \text{SOA}_{\text{meas}}$, which is the wall-deposition-corrected SOA mass. The wall-deposition correc-
tion assumes that once a particle deposits on the wall, suspended gas-phase molecules no longer condense onto it; its growth
ceases. This corresponds to the technical assumption that $\omega = 0$, where $\omega$ is a proportionality factor that describes the degree
to which vapor condenses onto particles already deposited on the chamber walls compared to those suspended in the bulk of
the chamber: if $\omega = 0$, once a particle deposits on the chamber wall it is lost to the system and no longer acts as a condensation
sink; if $\omega = 1$, a particle deposited on the chamber wall acts as a condensation sink identically to that of a suspended particle
(Trump et al., 2016; Weitkamp et al., 2007). The SOA yield is bounded by the assumptions that $\omega = 0$ and $\omega = 1$. The extent of
difference between these cases is dependent on characteristics of the chamber (e.g., the rate of particle-wall-deposition) and of
the chemical system (e.g., the amount of kinetic vs. equilibrium particle growth that occurs) (Trump et al., 2016).

To estimate the upper bound ($\omega = 1$) of the yield, we assumed that only particles that deposited after the onset of oxidation
would take up vapor. That is, inorganic seed deposited during the background collection period of each experiment is not
considered.

While different-sized particles both deposit to the wall at different rates and grow due to condensation at different rates, to
simplify the calculation of the SOA yield upper bound, the volume-weighted mean diameter of the suspended size distribution
was determined for each time point such that $D_{p,av,t} = \left( \frac{1}{N_{total,t}} \sum_{i=1}^{nbins} \left( D_{p,i}^3 N_{i,t} \right) \right)^{1/3}$, where $N_{total,t}$ is the total number
concentration at time point $t$, $nbins$ is the number of diameter size bins measured by the SMPS, $D_{p,i}$ is the mean diameter of





each size bin, and $N_{i,t}$ is the number concentration of particles of diameter $D_{p,i}$ at time $t$. Then, the upper bound assumption of SOA mass formed during the experiment is given by

$$\Delta\text{SOA}_{\text{meas},\omega=1} = \Delta\text{SOA}_{\text{meas}} + \frac{\pi}{6}\rho\sum_{t=t_1}^{t_{end}}\left[\left(D^3_{p,av,t_{end}} - D^3_{p,av,t}\right)N_{lost,t}\right] \tag{2}$$

where $\rho$ is the particle density, $N_{lost,t}$ is the number concentration of particles lost to the chamber wall between $t_i$ and $t_{i+1}$, and $t_{end}$ is the time in the experiment considered. This calculation was performed for 1 min time steps.

Table 1 shows the SOA yields calculated with uncertainties for the $\omega = 0$ and the $\omega = 1$ assumption. The SOA yield calculation with both $\omega = 0$ and $\omega = 1$ is shown for experiment R1 in Fig. 1. Since the difference between the SOA yield calculated with $\omega = 1$ and with $\omega = 0$ is dependent on the amount of organic aerosol that deposits onto the chamber walls, experiments with a higher initial aerosol concentration or that simply last for a longer period tend to have a greater disparity between SOA yields calculated with the $\omega = 0$ assumption and those calculated with the $\omega = 1$ assumption. Even so, for all the experiments

considered here, the $\omega = 1$ calculated SOA yield is within the uncertainty of the SOA yield found assuming that $\omega = 0$.

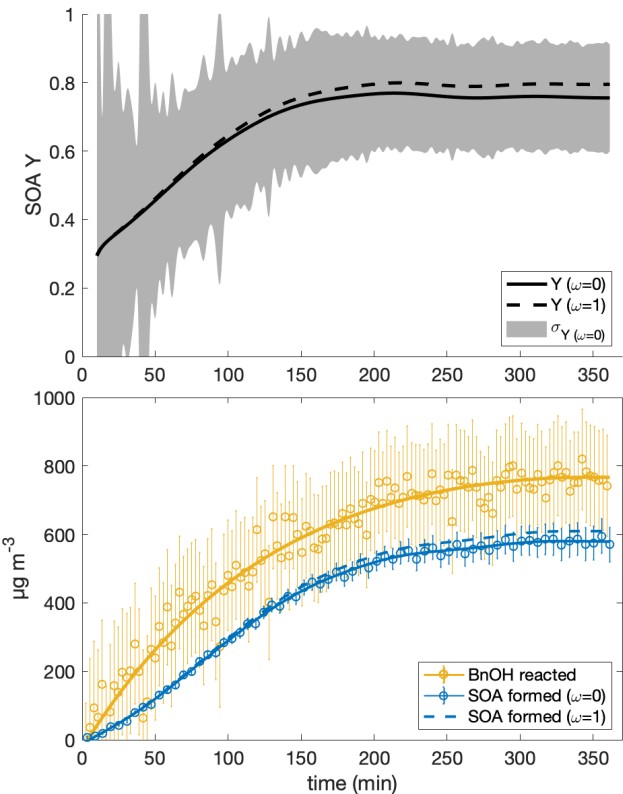

**Figure 1.** (a) The SOA yield for experiment R1 calculated with the assumption that $\omega = 0$ is shown as a solid curve and with $\omega = 1$ as a dashed one. The shaded regions is the associated uncertainty for the $\omega = 0$ case. Panel (b) shows the wall-deposition-corrected mass concentration of SOA formed assuming $\omega = 0$ (blue solid curve fitted to the circles and error bars) and $\omega = 1$ (dashed blue curve). The measured mass concentration of benzyl alcohol is the yellow circles with associated error bars, to which the yellow curve is fit.

## 3.2 Corrections

The chamber walls have, primarily, two effects on the SOA yield results: particles with organic mass on them may deposit on the chamber walls and not be detected (called particle wall deposition) or low-volatility compounds that, in the atmosphere, would condense onto suspended particles and form secondary organic aerosol mass instead deposit directly onto the chamber

walls (called vapor wall deposition).

Since vapor wall deposition involves the loss to the wall of not just the precursor compound, in this case benzyl alcohol, but also of all the oxidation products, which, as is the case here, are often not all fully measured and characterized, it is difficult to directly correct for the effect of vapor wall deposition on the observed SOA yield. Instead, one can minimize its effect by increasing the presence of the suspended aerosol surface area concentration so that the suspended aerosol outcompetes the

chamber wall as a condensation sink. To do so, however, increases the effect of particle wall deposition because as there are





more particles in the chamber, a greater fraction will generally deposit onto the chamber walls (due to a nonlinear decay) (Charan et al., 2019).

Noting that one must always account for particle wall deposition, since even a nucleation experiment will produce particles that may deposit on the chamber walls while one is attempting to measure them, we take this approach of correcting for particle

wall deposition and operating our experiments in a regime that minimizes the effect of vapor wall deposition.

### 3.2.1 Particle-wall deposition

To determine the particle-wall-deposition correction parameters for the 17.9 m$^3$ chamber, two-parameter fits to the eddy-diffusivity coefficient ($k_e$) and the mean electric field experienced within the chamber ($\bar{E}$), as outlined in Charan et al. (2018), were performed on dry, ammonium sulfate experiments with an assumed density of 1770 kg m$^{-3}$. Two experiments were

carried out for 8 h in the dark with only ammonium sulfate seed present, one was a 6 h experiment under irradiation, and an additional four were 4 h dark experiments with the precursors of a VOC oxidation experiment. All dark experiments were carried out at 25.6°C and that in the presence of light was performed at 28.6°C. Analysis began 30 min after initial mixing and used 15 size bins to improve the counting statistics. All bins were included in analysis.

When a two-parameter minimization on $k_e$ and $\bar{E}$ for each experiment was performed following the protocol described in

Charan et al. (2019), initial guesses of $k_e$ were varied between 0.15 and 5 s$^{-1}$ and of $\bar{E}$ between 0 and 50 V cm$^{-1}$. Three of the seven experiments gave $\bar{E} < 0.1 \times 10^{-9}$ V cm$^{-1}$, and the other four gave $\bar{E}$ = 2.1, 2.3, 3.9, and 5.1 V cm$^{-1}$. When all the experiments were analyzed together, with an initial guess of $k_e$ varying between 0.001 and 10 s$^{-1}$, the minimization function converged with $k_e$ = 0.0769 s$^{-1}$. Even for those experiments that gave $\bar{E} \neq 0$ when optimized, all fit approximately as well to their one-parameter minimization and to the all-experiment optimized value ($k_e$ = 0.0769 s$^{-1}$) as to their individually optimized

values. One-parameter optimization (optimizing only for $k_e$, while assuming $\bar{E}$ = 0) was also performed for each of the 7 experiments. Uncertainty in wall-loss was determined by taking the smallest $k_e$ value found from each of these experiments (0.0004 s$^{-1}$) as a lower bound and the largest $k_e$ value (0.5 s$^{-1}$) as an upper bound. The total mass concentration of SOA formed, which was used to calculate the SOA yield, was found from a smoothing spline fit of the particle-wall-deposition-corrected volume concentration (R$^2 \geq 0.994$). Wang et al. (2018a) have shown, for a similarly configured chamber to those

used here, that neither UV lights turning on and off, nor flushing of the chamber, nor gas-phase injections had an effect on particle wall deposition.

As additional verification, for three experiments performed under the standard replication conditions, the contents of the chamber were allowed to sit undisturbed for 4 h prior to the lights being turned on. During these 4 h, the wall loss correction was performed using the parameters $k_e$ = 0.0769 s$^{-1}$ and $\bar{E}$ = 0, for which it was verified that these values gave constant volume

concentrations.

Prior to the commencement of oxidation, all experiments were mixed and then allowed to sit undisturbed for $\geq 1$ h. During this background-collection period, during which we assume no aerosol growth took place, the wall-deposition-corrected volume concentration was calculated using the $k_e$ and $\bar{E}$ parameters given above. To quantify the degree to which this volume concentration was properly wall-deposition corrected, the slope of a linear fit of the volume concentration as a function





of the time (with a 95% confidence interval) during this background period is reported in Table 1. Since experiment S1 was performed in the absence of initial seed, the aerosol volume concentration during the background collection time was 0 and no slope is reported. For all 20 experiments in which a SOA yield is reported (excluding S1), the wall-deposition-corrected volume concentration during the background collection time was relatively constant: the absolute value of the slopes for all experiments was $< 0.1$ $\mu m^3$ $cm^3$ $s^{-1}$ and the mean was 0.03 $\mu m^3$ $cm^{-3}$ $s^{-1}$.

The initial particle surface area concentration was taken to be the average of the wall-loss corrected values of the seed volume during the background-collection period.

### 3.2.2 Vapor-wall deposition

Based on three periods of vapor wall loss prior to experiment S3, each >100 min, the timescale of the loss of benzyl alcohol to the Teflon chamber walls is on the order of days (~2 to 5 days). While benzyl alcohol itself may be lost slowly, the significant

SOA yield dependence on initial seed surface area seen for the similar toluene-oxidation system (Zhang et al., 2014) suggests that other benzyl alcohol oxidation products might partition to the wall. A low derived accommodation coefficient of vapor to suspended particles ($\alpha_p$), as discussed in Sect. 6.2, also implies the presence of a seed surface area effect. For, the slower the gas-particle equilibration, the more likely that the chamber wall is an attractive condensation sink.

To understand the extent to which the chamber wall is competitive with the suspended aerosol as a condensation sink, the

initial seed surface area concentration was varied for otherwise identical experimental conditions. Figure 2 shows this observed SOA yield, where no vapor-wall-deposition corrections are performed, for a range of initial seed surface area concentrations. Above ~1800 $\mu m^2$ $cm^{-3}$, there appears to be little change in the observed SOA yield; thus, we assume that the effect of vapor wall deposition is minimal.





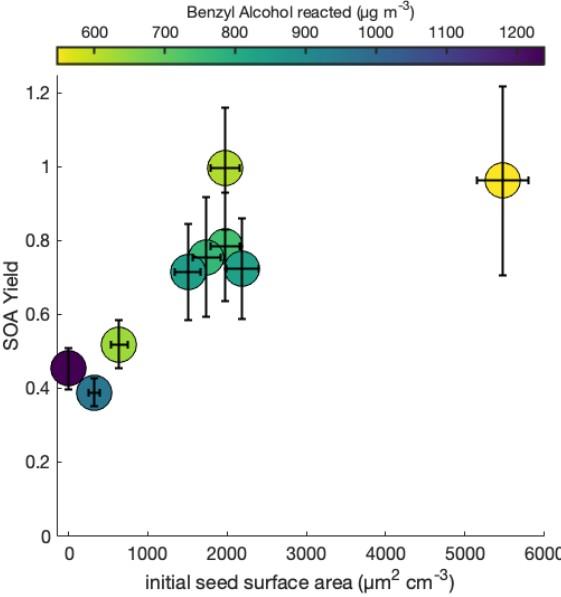

**Figure 2.** Variation in observed benzyl alcohol SOA yield with an initial NO mixing ratio of 80 ppb at 291 K as a function of the amount of benzyl alcohol reacted and the initial aerosol seed surface area. The lack of a difference in the yield over differing seed surface areas above ~1800 $\mu m^2$ $cm^{-3}$ indicates that the experiments lie within a regime where the seed surface area does not affect the measured SOA yield.

For each chamber and each chemical system, the initial seed surface area concentration at which the effect of vapor wall deposition is no longer significant is different: this is a function of, among other factors, the particle-vapor equilibration time, the accommodation coefficient of the gas-phase product to the chamber walls, the chamber dimensions, and the initial precursor concentration (Charan et al., 2019; Zhang et al., 2015).

In theory, the fact that we can neglect the effects of vapor wall deposition on SOA yield at a temperature of 291 K and an initial NO mixing ratio of ~80 ppb (as is the case for experiments R1–5 and S1–4, which are shown in Fig. 2), does not mean that we can neglect the effects for all temperatures and all NO mixing ratios, since different experimental conditions may change the chemistry of the system. However, while the identities and relative ratios of gas-phase products may differ for the different experiments explored in this paper, and hence the propensity to partition into the wall may vary, it is assumed that the products are sufficiently similar that the range at which vapor-wall deposition is considered insignificant remains the same. And, so, we apply the assumption that vapor wall deposition minimally affects the observed SOA yield at initial seed surface area concentrations above ~ 1800 $\mu m^2$ $cm^{-3}$ to all experiments in this paper.

### 3.2.3 Effect of corrections on measured SOA yield

The SOA yield is defined as the ratio of the mass of aerosol formed to the mass of precursor reacted (see Eq. 1). One may overestimate the yield by underestimating the amount of benzyl alcohol reacted or by overestimating the amount of aerosol formed. If the particle-wall-deposition adjustment overcorrects the aerosol formed, it would seem as if a higher yield exists





than that in actuality. Table 2 shows the SOA yield that would be calculated assuming that no particles were lost to the chamber walls during the experiment. Except for experiment R3 and L1, which ran for 12 h and 17 h, respectively, the raw particle volumes at the end of the experiments were > 80% of the wall-deposition-corrected volumes. So, even if there are errors in the particle-wall-deposition correction, the SOA yields will still be quite large.

**Table 2.** SOA yields in the absence of particle-wall-deposition corrections. Values are given assuming $\omega = 0$. The number in parentheses is the percent of the SOA yield (assuming $\omega = 0$) without accounting for particle wall deposition compared to with accounting for it.

| Label/Day | SOA Y | SOA Y (no correction) |
|---|---|---|
| R1/190321 | 0.76±0.16 | 0.68 (89%) |
| R2/190323 | 0.99±0.16 | 0.87 (88%) |
| R3/190312 | 0.70±0.13 | 0.54 (77%) |
| R4/190319 | 0.79±0.15 | 0.70 (88%) |
| R5/190128 | 0.72±0.13 | 0.58 (81%) |
| S1/191219 | 0.45±0.06 | 0.41 (91%) |
| S2/191002 | 0.39±0.04 | 0.34 (87%) |
| S3/190930 | 0.52±0.06 | 0.48 (92%) |
| S4/190325 | 0.96±0.25 | 0.81 (84%) |
| T1/190419 | 0.60±0.11 | 0.54 (89%) |
| T2/190417 | 0.54±0.09 | 0.48 (88%) |
| T3/190422 | 0.63±0.13 | 0.53 (84%) |
| T4/190410 | 0.37±0.08 | 0.32 (87%) |
| N1/190408 | 0.70±0.12 | 0.63 (91%) |
| N2/190403 | 0.68±0.16 | 0.61 (90%) |
| N3/190426 | 0.66±0.17 | 0.56 (84%) |
| N4/190401 | 0.60±0.09 | 0.54 (90%) |
| N5/190424 | 0.54±0.10 | 0.46 (85%) |
| N6/190405 | 0.47±0.08 | 0.42 (89%) |
| E1/200109 | 0.35±0.05 | 0.29 (82%) |
| L1/190110 | 0.37±0.18 | 0.10 (27%) |

Vapor-wall deposition will only decrease the observed mass of aerosol formed. If experiments were not run at a sufficiently
large aerosol surface area concentration to neglect the loss of gas-phase products to the chamber walls, the true SOA yield will only be larger than what is reported here.





## 4   Aerosol chemical composition

Throughout all the experiments, the O:C ratio also first decreases and then increases. Figures 3 and 4 show the aerosol chemical composition analyzed at different temperatures and NO mixing ratios, respectively. If particle growth is mass-transfer limited

(supported by a modeled $\alpha_p \sim 10^{-2}$, see Sect. 6.2), this might simply be a result of the greater abundance of higher volatility oxidation products at the beginning of the experiment. Only the lowest volatility (which are, presumably, compounds with the highest O:C ratios) condense initially, but as higher volatility compounds build up they may eventually partition into the aerosol phase, decreasing the O:C ratio. As lower volatility second- and third-generation compounds are formed, these might then increase the O:C ratio observed. There may also be particle-phase chemical reactions occurring that leads to the change in

O:C ratio throughout the experiment or the observed change could result from a change in the nitrogen-containing compounds in the aerosol-phase. Note that, when there is a large contribution of organonitrates to the aerosol, the O:C ratio will be an underestimate (Aiken et al., 2008).





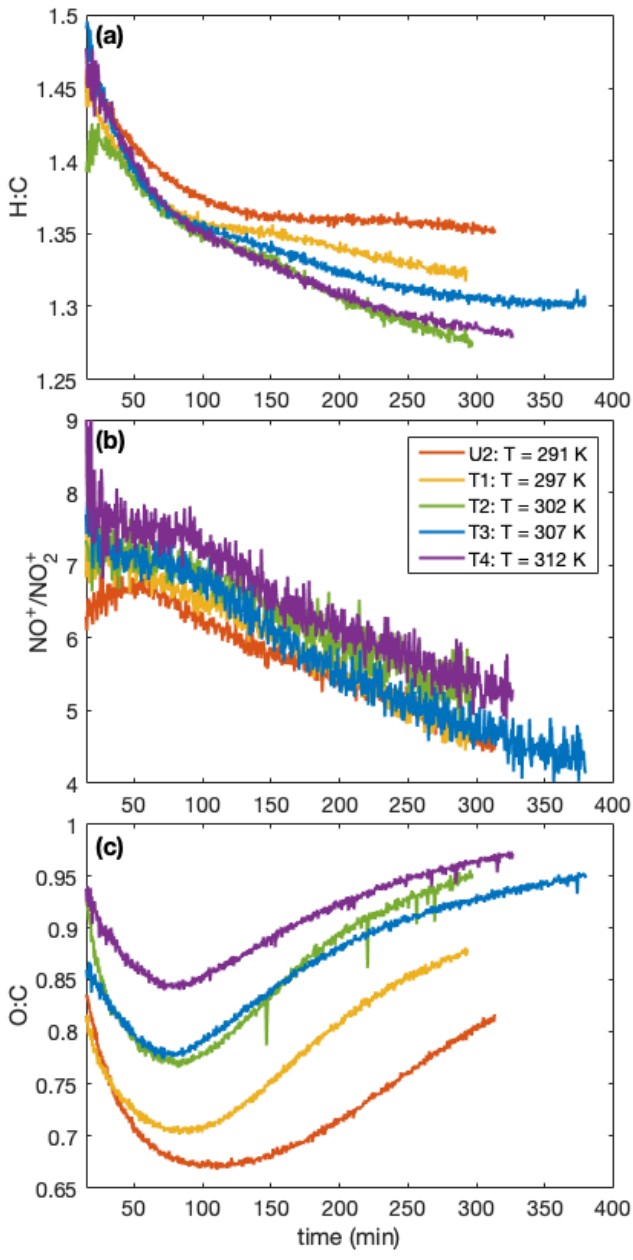

**Figure 3.** Variation in (a) the hydrogen to carbon atomic ratio, (b) the $NO_x^+$ ratio, and (c) the oxygen to carbon atomic ratio indicate that the difference in SOA yield observed at different temperatures might be a result of chemical differences in the aerosol formed. Absolute uncertainties are 13% and 28% for the H:C and O:C ratios, respectively. Since the ratios are relevant only when there is a sufficient amount of aerosol present, the first 15 min after oxidation are not shown. A SOA yield is not calculated for experiment U2 due to uncertainties in the rate of particle-wall deposition, but that should not affect the chemical composition of the aerosol.




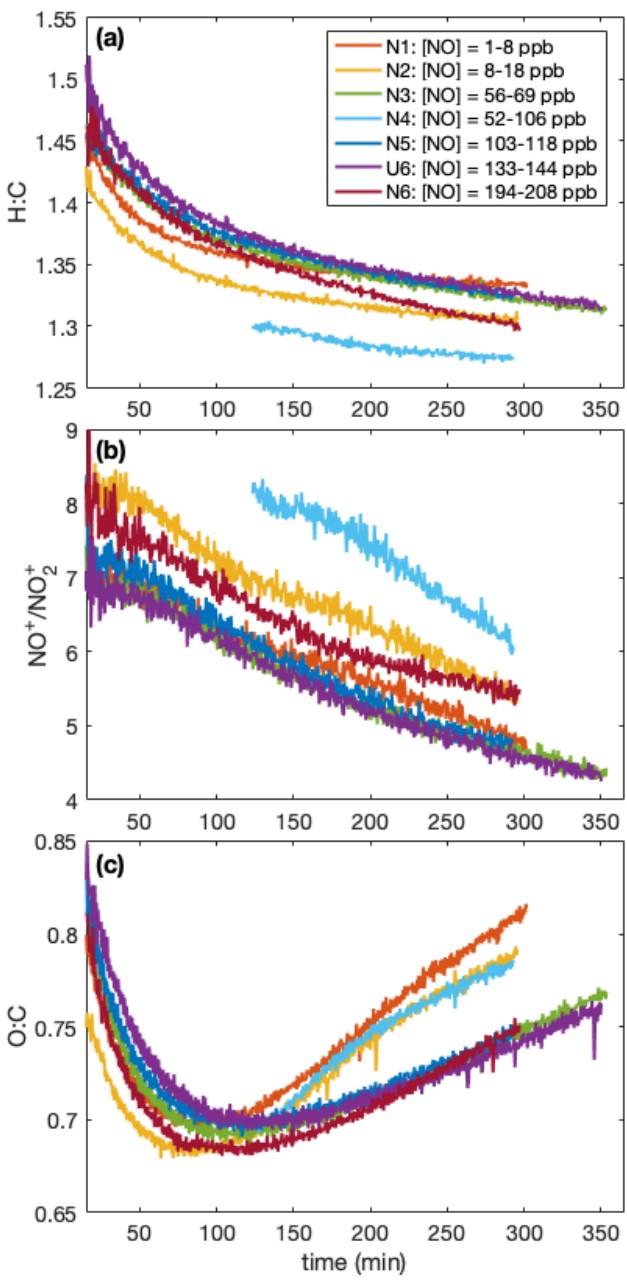

**Figure 4.** Variation in the (a) hydrogen to carbon atomic ratio, the (b) NO to NO$_2$ signal mass ratio, and the (c) oxygen to carbon atomic ratio indicate that the difference in SOA yield observed at different NO mixing ratios is a result of chemical differences in the aerosol formed. Absolute uncertainties are 13% and 28% for the H:C and O:C ratios, respectively. Since the ratios are relevant only when there is a sufficient amount of aerosol present, the first 15 min after oxidation are not shown. Data were collected only after ~2 h of oxidation for experiment N4. A SOA yield is not calculated for experiment U6 due to uncertainties in the rate of particle-wall deposition, but that should not affect the chemical composition of the aerosol.



It appears that at the beginning of each experiment, the first secondary organic aerosol formed comprised a significant portion of organonitrates (as much >20% by mass), as shown in Fig. A1. While the mass fraction of organonitrates is not reported for

the experiments shown in Figs. 3 and 4 (due to calibration issues), the $NO_x^+$ ratio trend is the same as that for the experiments shown in Fig. A1, where the mass fraction can be reported. Note that one pathway to form organonitrates is by reaction with the nitrate radical; since all our analysis from the AMS is of experiments with the ultraviolet lights on, one does not expect a significant concentration of nitrate radicals (Seinfeld and Pandis, 2016). Instead, we expect the organonitrates to have been formed by a $RO_2 \cdot + NO$ reaction; this reaction has a high gas-phase yield for organonitrates for large compounds (Arey et al.,

2001; Rollins et al., 2010). As oxidation continued, more non-nitrogenated organic compounds condensed into the particle phase decreasing the mass concentration of organonitrates. Simultaneously, the $NO_x^+$ ratio decreased, which could have been caused by nitric acid, formed from $OH + NO_2$, partitioning into the aerosol phase and forming nitrate ions. Partitioning of $HNO_3$ into secondary organic aerosol has been observed by Ranney and Ziemann (2016). Another possibility is that other compounds, such as organonitrites, might produce $NO_2^+$ fragments that lower the $NO_x^+$ ratio throughout the experiment.

Indeed, UPLC analysis found a high prevalence of compounds of the form $RNO_2$ (see Table A1), which likely will not lead to the same $NO_x^+$ ratios as organonitrates and might contribute $NO_2^+$ fragments that could lower the $NO_x^+$ ratio. For all experiments with filters collected (N1–3 and U1–6), nearly all compounds detected with UPLC analysis were nitroaromatics. This indicates that the low-volatility products that condense into the aerosol phase retain their aromatic rings. It is possible, however, that there are non-ring retaining compounds which condense onto SOA that are simply not detectable by the UPLC.

Some of the ring-retaining compounds have $C_7$ structures, as does benzyl alcohol. However, several of the compounds detected are $C_6$ structures, indicating the possible loss of the methanol group. In particular, UPLC analysis showed a particularly high concentration of nitrocatechol in the aerosol. The atomic ratios of oxygen to carbon atoms (O:C) are quite large: between 0.6 and 1.0, which matches that of very oxygenated rings, but could also match nitrocatechol (O:C of 0.67).

The prevalence of nitroaromatics may be because the UPLC analysis method is particularly sensitive to nitroaromatics: the

detection of aerosol phase compounds via the UPLC/MS method is limited to detecting compounds that are water soluble and lie within the detection limits of the instrument. Though filter samples were stored at low temperatures, on-filter chemistry may be possible. Certain compounds may also be prone to hydrolysis when in the aqueous phase, which may alter the molecular weight of the original compounds collected in the particle phase (Zhang et al., 2016).

Nevertheless, it is clear that there are many nitrogen containing compounds in the particle phase. Differences in aerosol

chemical composition as a function of temperature and NO concentration is discussed in Sects. 5.2 and 5.3.

## 5 SOA yields

### 5.1 Time dependence

While, usually, the SOA yield is reported as a single number at the end of an experiment, it can also be understood as a function of time since multiple generations of oxidation products usually exist (Cappa et al., 2013). For example, in the $\alpha$-pinene system,

the SOA yield has been shown to depend on the total hydroxyl radical exposure (Donahue et al., 2012; Wang et al., 2018b).





Figure 5 shows, for each experiment, the terminal SOA yield and the bands indicating at which times each of the experiments lie within 10%, 5%, and 1% of the final reported yield. The most atmospherically representative SOA yield is that to which the experiments converge. For almost all the experiments, the yields appear to have converged sufficiently to justify the reporting of the final yield, though the benzyl alcohol concentration may not yet have all reacted (see Fig. 6); as more reacts, more aerosol is

formed but the SOA yield levels out. Experiments R3 and R5, which were run for considerably longer than other experiments, show that the final SOA yield changed little from earlier in oxidation, when the other experiments were terminated. Instead of looking at this in terms of reaction time, one can see instead the SOA yield as a function of the amount of the initial benzyl alcohol reacted ( Fig. 7) and see that the yield also converges in terms of the fraction of benzyl alcohol reacted.

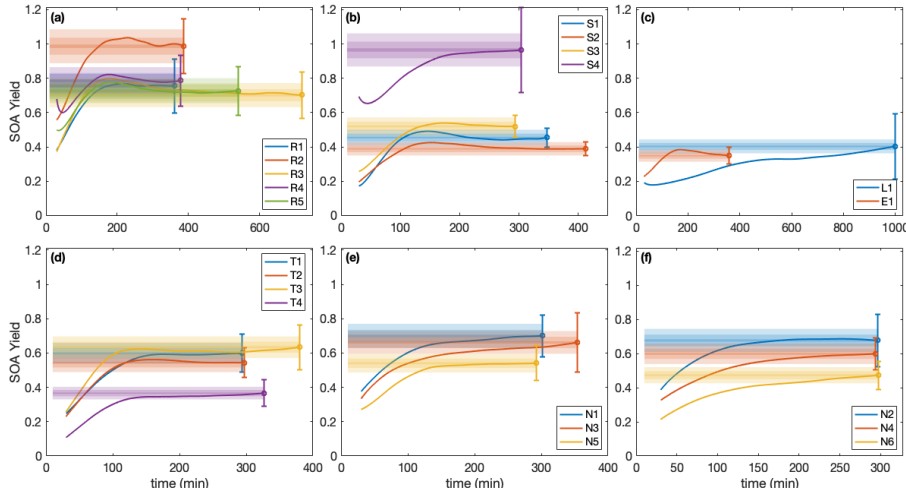

**Figure 5.** SOA yield calculated assuming $\omega = 0$ as a function of time for (a) reproduction experiments, (b) different initial surface area experiments, (c) the low light strength experiment (L1) and the initial $NO_2$ experiment (E1), (d) different temperature experiments, and (e–f) variable constant NO mixing ratio experiments. The measured SOA yields are the solid line and the reported end yield is the circle with the reported error bars. The lightest shaded region is ±10% of the reported end yield, the medium-shared region is ±5%, and the darkest shaded region is ±1%.




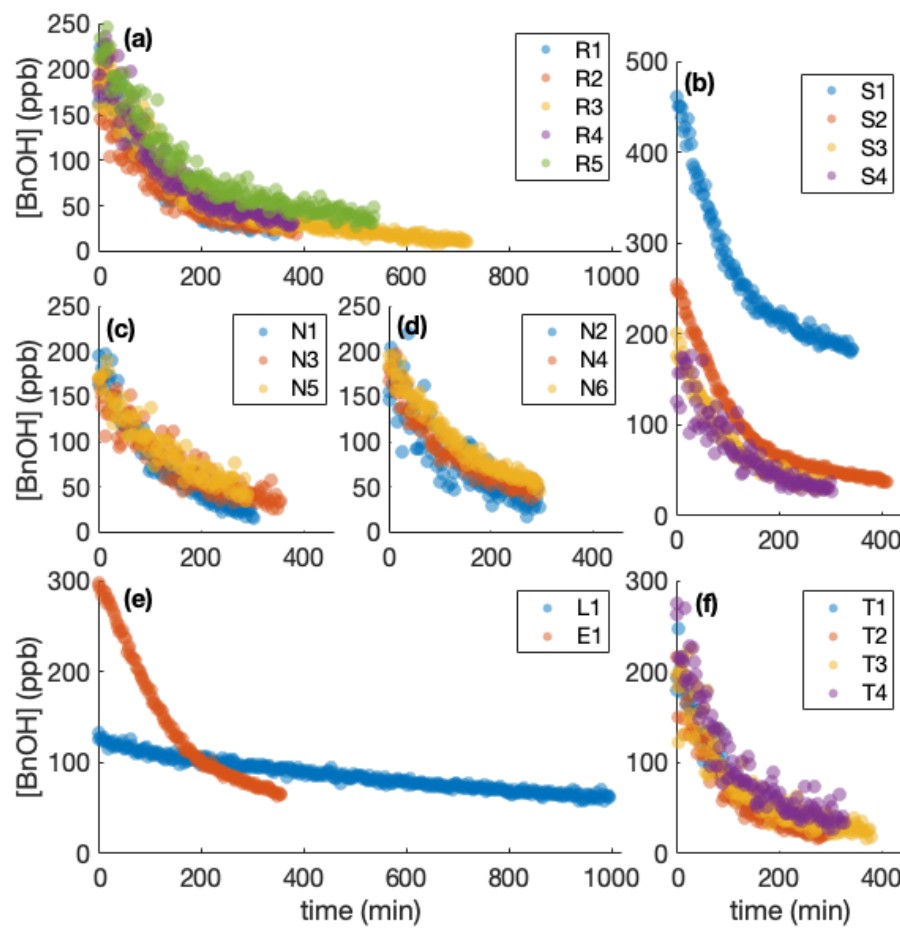

**Figure 6.** Benzyl alcohol decay for the (a) reproduction experiments, (b) different initial surface are experiments, (c–d) variable NO mixing ratio experiments, (e) the low light strength experiment (L1) and the initial $NO_2$ experiment (E1), and (f) the different temperature experiments. All panels are scaled the same in both axes. The x-axis is time since the commencement of oxidation. Except for experiment L1, which was run at ~10% the light strength of the other experiments, all experiments follow a similar decay curve.





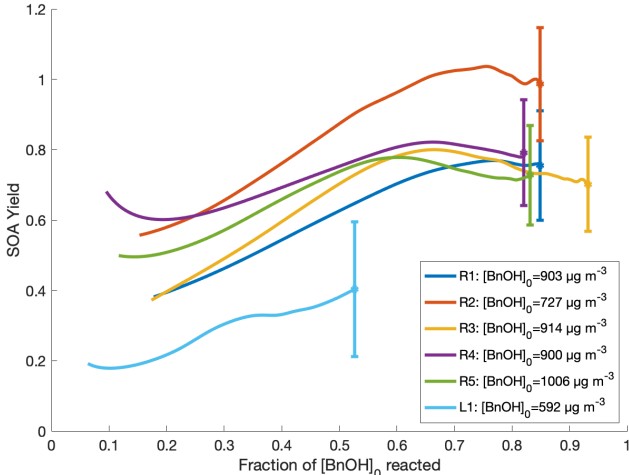

**Figure 7.** Secondary organic aerosol yields as a function of the fraction of initial benzyl alcohol reacted for experiments R1–5 and L1. All experiments were run under approximately the same conditions, although experiment L1 had a light strength of <10% of the other experiments.

Note that for experiment L1, also run for considerably longer than the other experiments, the light strength was $< 10\%$ of that in all the other experiments. At this lower oxidation rate, the SOA yield takes much longer to converge but does appear to be somewhat a function of the fraction of benzyl alcohol that has reacted at any given time. This shows that the convergence time depends on the rate of oxidation. Figure 6 shows the decay of benzyl alcohol throughout the experiment: for all except experiment L1, the curves show a very similar decay. The first-order exponential decay constant ($k_{\mathrm{BnOH+OH}}[\mathrm{OH}]$) for each experiment is given in Table 1.

## 5.2 Temperature dependence

Figure 8 shows the SOA yield of benzyl alcohol over a range of temperatures, all corresponding to approximately the same initial surface area range (1500–2800 $\mu\mathrm{m}^2\,\mathrm{cm}^{-3}$) and the same initial NO mixing ratio of ~ 80 ppb (see R1–5 and T1–4 in Table 1). In general, a lower yield of benzyl alcohol exists at higher temperatures; this is expected due to the decreased volatility of oxidation products at lower temperatures and to the increased rapidity of second-generation reactions, which may potentially form high volatility fragments before the lower volatility first-generation products have time to partition into the particle phase.



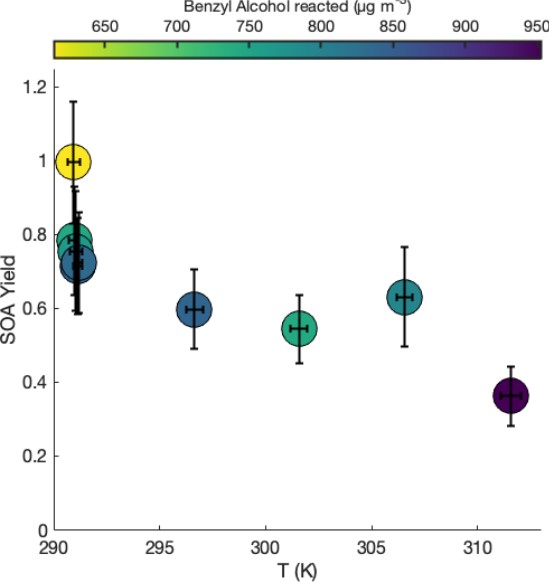

**Figure 8.** Variation in SOA yield over several hours of benzyl alcohol oxidation as a function of temperature with an initial NO mixing ratio of 72 to 81 ppb as a function of the amount of benzyl alcohol reacted for experiments R1–3 and T1–4. The color is proportional to the amount of benzyl alcohol that has reacted at the end of the experiment. Experiments began with between 78 and 102 ppb of benzyl alcohol and initial seed surface area concentrations of 1800 to 2900 $\mu m^2$ $cm^{-3}$. Error bars are given for the yields at the end of each experiment (experiment lengths are given in Table 1).

At the lowest temperature measured, where one would expect the greatest seed surface area effect (that is, the most competition between the wall and suspended aerosol condensation sinks), we have already determined that we are outside the range of the seed surface area effect (Fig. 2). So, one would not expect that the difference in SOA yield is related to competition with the chamber wall.

A higher SOA yield at lower temperatures is also supported by Fig. 3, which shows how the chemical makeup of the aerosol is different for aerosol formed at different temperatures: the O:C ratio is higher and the H:C ratio is lower on aerosol formed at higher temperatures, meaning that more volatile compounds that might condense at lower temperatures (and have a smaller O:C ratio and a lower H:C ratio) do not condense at the higher temperature (panels a and c). Though the difference is slight, there is a trend for a larger $NOx^+$ ratio (panel b) and, correspondingly, a larger mass fraction of organonitrates at higher temperatures.

The former indicates that the organonitrates may be less volatile than other nitrogen-containing compounds that may condense into the aerosol phase (including, potentially, inorganic ammonium nitrate). The latter suggests that the gas-phase branching may be different. It may be that fewer organonitrates are formed at lower temperatures.





### 5.3 Nitric oxide mixing ratio dependence

To probe the different chemical pathways that form, the SOA yield dependence on variable NO concentrations was investigated

(Fig. 9). NO mixing ratios were maintained throughout experiments N1–6 and U6, leading to an increase in the total $NO_x$ in the system. $NO_x$ increased by ~60 ppb for experiment N1 and ~100–200 ppb for experiments N2–6 and U6. Generally, the SOA yield seems to decrease with increased NO concentration.

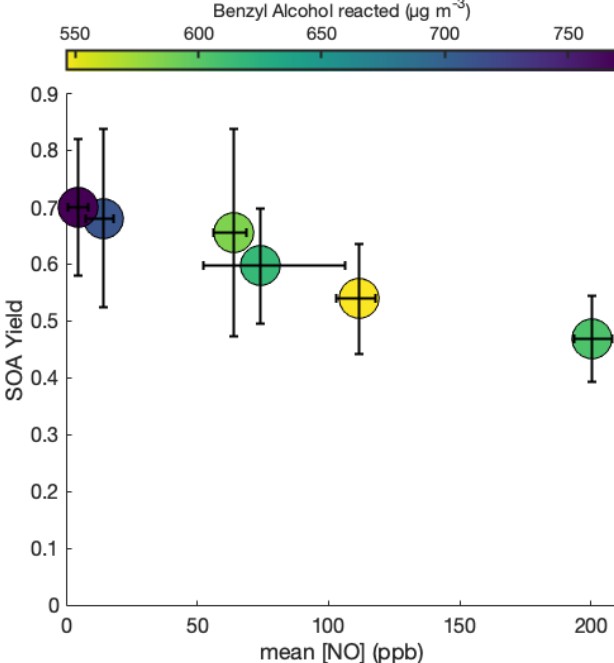

**Figure 9.** SOA yield under different constant NO conditions for experiments N1–6. All experiments were performed at 291 K, with initial benzyl alcohol mixing ratios between 70 and 82 ppb, and with initial seed surface area concentrations of 1800 to 2900 $\mu m^2\ cm^{-3}$. The x-axis error bars show the full range of NO concentrations experienced throughout the experiment.

As shown in Fig. 4c, there are also larger O:C ratios after ~2 h of oxidation for the lower NO mixing ratios (N1, N2, and N4). Note that experiment N4 appears to behave more similarly to N1–2 than to N5–6 and U6; the control on the NO mixing

ratio for N4 was much less successful than for the other constant NO experiments (see the error bars in Fig. 9). While the [NO] throughout experiment N4 was, on average, 74 ppb, it was only 62 ppb on average during the first 3 h of oxidation (experiment N3 had an average [NO] of 62 ppb during the first 3 h of oxidation).

We suspect that there are a large number of nitroaromatics in the organic aerosol (see Sect. 4). Perhaps at higher NO concentrations there are more nitroaromatics, and these compounds are more volatile than the nitrogen-free oxidation products (such

as the very oxygenated rings). Though the differences in H:C and O:C ratios are slight, the larger O:C ratios—corresponding to the very oxygenated rings—that are seen at lower NO concentrations support the theory that the compounds formed differ (see Fig. 4).





Experiment E1, which is similar to experiments R1–5 except that, prior to the beginning of oxidation, it begins with 71.0±0.8 ppb of $NO_2$ and no NO, shows a much lower SOA yield than that from experiments R1–5. This suggests that it is the NO

that is the relevant reactant that causes initially high SOA formation. This is supported by the significant mass fraction of organonitrates at the beginning of the experiments; organonitrates are formed by $RO_2\cdot$ reaction with NO.

## 6  Benzyl alcohol oxidation chemistry

### 6.1  Theory

Oxidation of benzyl alcohol in the present system occurs predominantly via reaction with the hydroxyl radical (OH). The reac-

tion with OH proceeds via H-abstraction from the $CH_2$ group or OH addition to the aromatic ring; its products are hypothesized to include benzaldehyde, 3-hydroxy-2-oxopropanal, butenedial, and glyoxal (Wang, 2015). Measured rate constants (Harrison and Wells, 2009; Bernard et al., 2013) for reaction with the OH radical found using a relative-rate method are $(2.8\pm0.4)\times10^{-11}$ $cm^3$ molecule$^{-1}$ s$^{-1}$ at $297\pm3$ K.

A chemical understanding of the gas-phase oxidation of benzyl alcohol is useful for modeling the system, which can aid in

understanding the gas- and particle-phase dynamics. Note that while gas-phase dynamics affect the SOA formed, the assumptions made in this section do not affect the measured SOA yields and are only used for understanding the system.

The measured gas-phase yield of benzaldehyde from the reaction of benzyl alcohol with OH is 24±5% at 298 K (Harrison and Wells, 2009; Bernard et al., 2013), which also matches well with a theoretical value of 29.6% (Wang, 2015). For gas-phase modeling and related optimization (Sect. 6.2 and 6.3), we use branching ratios following the results of Wang (2015), which

combine theoretical and experimental branching results: 0.25 to form benzaldehyde, 0.11 to form $o$-hydroxy-benzyl alcohol (note that this differs somewhat from the measured yield of 0.22 Bernard et al. (2013)), 0.23 to high volatility fragments (including glyoxal and butanedial), and the remaining 0.41 to low volatility and ring-containing products. Since the intermediate reactions are theoretically much faster than the initial reaction of OH with benzyl alcohol (except for the reactions of benzaldehyde), we employ the mechanism given in Fig. 10, in which compounds of similar volatilities are grouped into the

precursor (BnOH), benzaldehyde (BnAl), fragments (Frags), very oxygenated rings (VORings), and hydroxy-benzyl alcohol (HOBnOH).







**Figure 10.** Benzyl alcohol reaction scheme used for simulations, roughly derived from Wang (2015).

Molecular weights used for each compound class are the weighted values by component (predicted by Wang (2015)) given in Table 3. For each compound class, the estimated vapor pressure is the component-weighted value found using the EVAP-ORATION method (Topping and Jones, 2016) (note that using EVAPORATION gives results similar to the Nannoonal and Myrdal methods) at the mean temperature of the experiment under consideration; for reference, the saturation mass concentration $C^*$ is given in Table 3 at 291 K. The Oxygen-to-Carbon ratio is also given for each compound class. Note that none of these predicted products are organonitrates or other nitrogen-containing organic compounds, as observed in the aerosol (see Sect. 4). The lack of nitrogen-containing products, especially at the very beginning of oxidation, could be responsible for some of the discrepancy between the observed and simulated results.

**Table 3.** Compound class properties for simulating chamber experiments.

| Compound Class | Abbreviation | MW (g mol$^{-1}$) | O:C | $\log_{10} C^*$ at 291 K (µg m$^{-3}$) | Initial Branching Ratio |
|---|---|---|---|---|---|
| benzyl alcohol | BnOH | 108.14 | 0.14 | 5.73 | |
| benzaldehyde | BnAl | 106.12 | 0.14 | 6.88 | 0.25 |
| fragments | Frags | 87.84 | 0.75 | 7.25 | 0.23 |
| very oxygenated rings | VORings | 188.13 | 0.86 | 2.13 | 0.41 |
| hydroxy-benzyl alcohol | HOBnOH | 124.13 | 0.29 | 5.79 | 0.11 |





## 6.2 Chamber simulation

All optimization procedures and modeling are based on a fixed-bin model, as described in Charan et al. (2019). A density of
1.4 g cm$^{-3}$, consistent with past work on similar compounds (Dommen et al., 2006; Kroll et al., 2005, 2006; Brégonzio-Rozier
et al., 2015), and a surface tension of 28.21 dyn cm$^{-1}$, that of benzene particles (Seinfeld and Pandis, 2016), are assumed
for the particles with SOA. Wall accommodation coefficients are calculated using the saturation mass concentrations of each
compound class (see Table 3) and the empirical fit described in Huang et al. (2018).

Modeling is carried out by fixing the decay of benzyl alcohol to the second-order exponential fit of the concentration. Since,
in theory, $\frac{d[\text{BnOH}]}{dt} = -k_{\text{OH}+\text{BnOH}}[\text{OH}][\text{BnOH}]$, if [OH] were constant throughout the experiment then [BnOH] should follow
a first-order exponential decay in time (the decay constant for this fit is given in Table 1). A slightly better fit was found to a
second-order exponential decay, so that fit is used for modeling.

Note that the model is not designed for nucleation experiments, because seeding the model with small particles requires
these particles to grow very quickly and, therefore, requires a much smaller time step. Hence, for the surface area experiments
we do not model experiment S1.

Because several of the simulation parameters are not precisely constrained (the equivalent saturation concentration of the
wall, $C_w$, the accommodation coefficient of vapor to suspended particles, $\alpha_p$, the accommodation coefficient of vapor to de-
posited particles, $\alpha_{pw}$, the accommodation coefficient of each product to the wall, $\alpha_{w,i}$), modeling of the system is associated
with considerable uncertainty. If one is confident in the branching ratios under each condition, then one could determine $\alpha_w$
for each product and optimize $\alpha_p$ and $C_w$ with the surface area and reproduction experiments (S2–4 and R1–4). Differences in
products could then be determined at different temperatures (using experiments T1–4) and at different constant NO concentra-
tions (using experiments N1–6).

With the base assumption that $\alpha_p = 1$, $\alpha_{pw} = 0$, and $C_w = 1 \times 10^4$ µg m$^3$, the model reproduces experiments R1–4 fairly well,
and most of the other experiments less successfully (see Fig. 11). Even for experiment R1, where the simulation captures the
total organic mass well (Fig. 11A), the size distribution evolution is less successfully captured (Fig. 12).





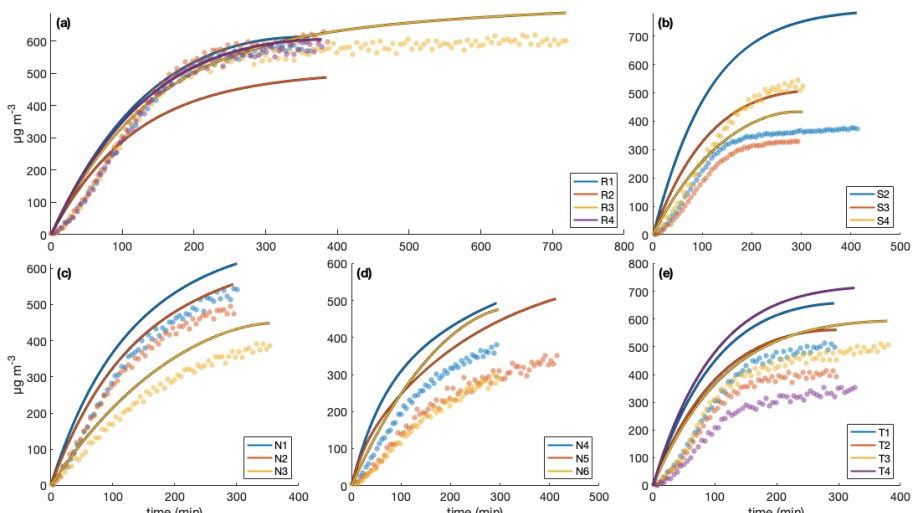

**Figure 11.** Comparison of measured (circles) and simulated (curves) secondary organic aerosol mass concentrations for different initial surface area concentrations assuming no vapor-wall deposition for the (a) reproduction experiments, (b) different surface area experiments, (c) low constant NO concentrations, (d) high constant NO concentrations, and (e) different temperature experiments. The decay of benzyl alcohol was simulated using a second-order exponential fit to the data. The accommodation coefficient of vapor to suspended particles $\alpha_p = 1$. Also, $\alpha_{pw} = 0$ and $C_w = 1 \times 10^4$ µg m$^3$. Simulation time steps were taken as 1 min.





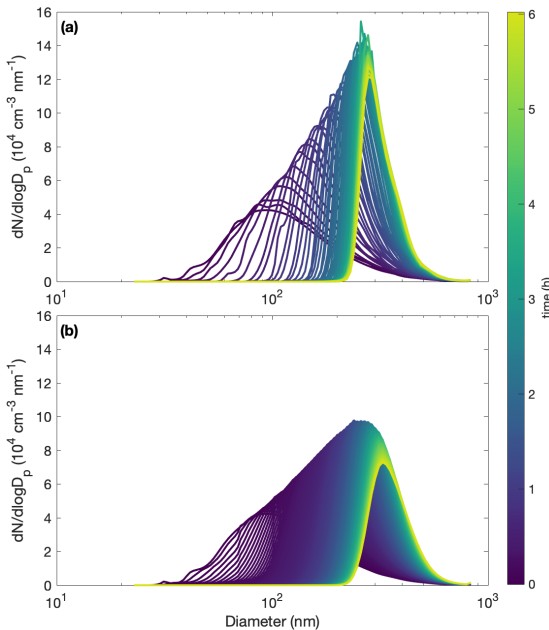

**Figure 12.** Comparison of measured (A) and simulated (B) particle size distributions throughout oxidation for experiment R1. The decay of benzyl alcohol is represented using a second-order exponential fit to the data. The accommodation coefficient of vapor to suspended particles $\alpha_p = 1$. Also, $\alpha_{pw} = 0$ and $C_w = 1 \times 10^4$ µg m$^3$. Computational time steps are taken as 1 min.

Deriving the true $\alpha_p$ by first optimizing solely for $\alpha_p$ (with $\alpha_{pw} = 0$ and $C_w = 10^4$ µg m$^{-3}$) for each experiment set (low NO mixing ratios, high NO mixing ratios, reproduction experiments, some surface area experiments with one reproduction experiment, and some surface area experiments with some reproduction experiments) shows that $\alpha_p$ is on the order of $10^{-2}$. This is the case for optimizations performed on all of the experiment sets. It is also the case if, instead of holding $\alpha_{pw}$ and $C_w$ at constant values, they are also allowed to change during optimization. These results are shown in Table 4. Note that this is less than the general average for many studied aerosol (~0.9) and specifically for the similar compound toluene, which was determined to be $0.3 \leq \alpha_p \leq 0.6$ (Liu et al., 2019).





**Table 4.** Optimization of parameters. The equivalent saturation mass concentration of the Teflon wall, $C_w$, has units of µg m$^{-3}$. The accommodation coefficient of vapor to suspended particles ($\alpha_p$) and of vapor to deposited particles ($\alpha_{pw}$) are unitless. For all optimizations, starting conditions were $\alpha_p = 1$, $\alpha_{pw} = 0$, and $C_w = 10^4$. When not optimized, $\alpha_{pw} = 0$, $C_w = 10^4$, and $\alpha_p$ is given in parentheses.

| Experiments Used for Optimization | $\alpha_p$ Optimized | $\alpha_p$ and $\alpha_{pw}$ Optimized | | $\alpha_p$, $\alpha_{pw}$, and $C_w$ Optimized | | | $C_w$ Optimized | | | $\alpha_p$ and $C_w$ Optimized | |
|---|---|---|---|---|---|---|---|---|---|---|---|
| | $\alpha_p$ | $\alpha_p$ | $\alpha_{pw}$ | $\alpha_p$ | $\alpha_{pw}$ | $C_w$ | $C_w$ ($\alpha_p = 1$) | $C_w$ ($\alpha_p = 10^{-1}$) | $C_w$ ($\alpha_p = 10^{-2}$) | $\alpha_p$ | $C_w$ |
| N1–3 | $2.2 \times 10^{-2}$ | $2.2 \times 10^{-2}$ | $6.2 \times 10^{-9}$ | $2.2 \times 10^{-2}$ | $4.0 \times 10^{-8}$ | $1.7 \times 10^4$ | $4.0 \times 10^8$ | $1.2 \times 10^8$ | $6.8 \times 10^2$ | $2.2 \times 10^{-2}$ | $1.9 \times 10^{-4}$ |
| N4–6 | $7.3 \times 10^{-3}$ | $7.3 \times 10^{-3}$ | $4.1 \times 10^{-9}$ | $7.5 \times 10^{-3}$ | $7.1 \times 10^{-9}$ | $1.7 \times 10^4$ | $2.0 \times 10^8$ | $2.8 \times 10^8$ | $2.2 \times 10^7$ | $7.6 \times 10^{-3}$ | $1.9 \times 10^4$ |
| R1–4 | $5.7 \times 10^{-2}$ | $5.7 \times 10^{-2}$ | $3.1 \times 10^{-8}$ | $6.0 \times 10^{-2}$ | $3.2 \times 10^{-8}$ | $1.7 \times 10^4$ | $7.9 \times 10^8$ | $2.9 \times 10^8$ | $1.5 \times 10^2$ | $6.0 \times 10^{-2}$ | $1.8 \times 10^4$ |
| S2–4 and R1 | $1.5 \times 10^{-2}$ | $1.5 \times 10^{-2}$ | $8.4 \times 10^{-9}$ | $1.5 \times 10^{-2}$ | $2.9 \times 10^{-8}$ | $1.7 \times 10^4$ | $8.2 \times 10^7$ | $6.6 \times 10^7$ | $2.3 \times 10^3$ | $1.5 \times 10^{-2}$ | $1.9 \times 10^4$ |
| S2–4 and R1–4 | $2.2 \times 10^{-2}$ | $2.2 \times 10^{-2}$ | $1.3 \times 10^{-8}$ | $2.2 \times 10^{-2}$ | $2.0 \times 10^{-8}$ | $1.7 \times 10^4$ | $6.6 \times 10^7$ | $6.6 \times 10^7$ | $7.5 \times 10^2$ | $2.3 \times 10^{-2}$ | $1.8 \times 10^4$ |

This suggests that mass-transfer limitations may be important for understanding the growth of SOA under these conditions. An accommodation coefficient close to 1 means that equilibrium between the gas- and particle-phase is quickly reached because there are few mass-transfer limitations. The smaller $\alpha_p$ found here indicates that the particles are highly viscous, i.e., that it takes some time for the particle-phase to equilibrate with the gas-phase. For systems with lower values of $\alpha_p$, one expects to see more of a seed surface area effect, which is discussed in Sect. 3.2.2.

Since any optimizations involving $\alpha_{pw}$ indicated very small values, for this chamber it appears that $\omega = 0$ is closer to reality than $\omega = 1$. This is because if $\alpha_{pw} \approx 0$, then effectively no gas-phase compounds are condensing onto particles that have already deposited on the chamber wall, which is the same as the assumption that $\omega \approx 0$.

## 6.3 Gas-phase insights

Benzaldehyde, which is a first-generation product of benzyl alcohol, photolyzes in addition to reacting with the OH radical (Bernard et al., 2013; Zhu and Cronin, 2000). Using absorption cross sections from the lamp-diode array from Thiault et al. (2004), assuming a quantum efficiency of 1, and normalizing the measured wavelengths in the chamber with the $j_{NO_2}$ value for the chamber gives $j_{BnAl} = 4.58 \times 10^{-4}$ s$^{-1}$.

Just as we employed chamber simulation to derive unknown chamber parameters, we can also determine which of the compound classes (Table 3) are most similar to benzaldehyde oxidation products. To do so, one must make assumptions about the other chamber parameters: here we take $\alpha_p = 1$, $\omega = 0$, $C_w = 10^4$ µg m$^{-3}$, the first-generation branching ratios given in Fig. 10, the assumption that photolysis products of benzaldehyde are volatile and act similarly to the Frags compound class, and the assumption that oxidation products of benzaldehyde condense into the particle phase (and so are most similar to the VORings compound class). This leaves only a single parameter to determine: $k_{OH+BnAl}[OH]$, which governs the amount of the oxidation product versus the photolysis product of benzaldehyde.

Performing a minimization on the difference between the predicted and measured secondary organic aerosol products while varying this parameter $k_{OH+BnAl}[OH]$ gives, for most of the experiments, a $k_{OH+BnAl}[OH] \approx 0$. Since we did not measure the benzaldehyde gas-phase concentration throughout the experiment, this result says nothing about the benzaldehyde that is





actually oxidized; it indicates, instead, that benzaldehyde does not form any condensable products. That is, it implies that the assumption that there might be any benzaldehyde products (from either photolysis or oxidation) that partition into the aerosol

phase is incorrect. We, therefore, assume that all benzaldehyde products become Frags, even those that are oxidation products. Since we are looking at the particle-phase results, and we assume that Frags do not condense onto particles, this is equivalent to assuming that $k_{\text{OH+BnAl}}[\text{OH}]=0$ without the constraint that no benzaldehyde reacts with OH. Experiments by Carter et al. (2005) also indicate that benzaldehyde oxidation products do not contribute significantly to the SOA formed from benzyl alcohol.

Depending on the temperature and the other experimental conditions (such as the NO mixing ratio), one would expect the chemistry to vary between experiments. The gas-phase concentration of hydroxy-benzyl alcohol (HOBnOH) has a molar mass of 124 g mol$^{-1}$ and is detected at M+19, corresponding to the addition of F$^{-}$ (Schwantes et al., 2017b). This signal normalized to the reactant ion signal by the initial benzyl alcohol concentration (expressed in signal normalized to reactant ion signal) for each of the experiments described here is given in Fig. 13. Note that this is, essentially, the HOBnOH concentration divided

by the initial benzyl alcohol concentration. The temporal evolution of HOBnOH for nearly identical experiments is fairly reproducible, as shown in panel a. The formation of HOBnOH or the rate at which it reacts away seem to increase slightly at higher temperatures (Fig. 13d) and possibly at higher constant NO concentrations (Fig. 13e but not 13f), but considering that the uncertainty in initial BnOH mixing ratio is on the order of 10% (see Table 1), it is difficult to make any concrete statements about the shift in gas-phase chemistry due to changing conditions except to say that changes are not hugely significant.





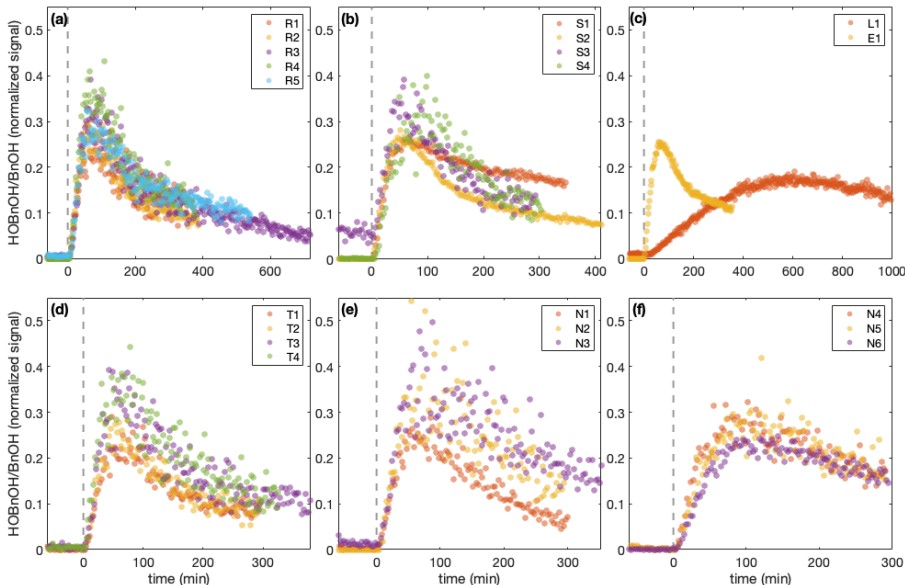

**Figure 13.** The normalized hydroxy-benzyl alcohol (HOBnOH) signal divided by the initial normalized benzyl alcohol signal (as calculated during the background collection period) for (a) reproduction experiments, R1–5, (b) different initial surface are experiments, S1–4, (c) the low light strength experiment, L1, and the initial $NO_2$ experiment, E1, (d) different temperature experiments, T1–4, (e) low constant NO mixing ratio experiments, N1–3, and (f) high constant NO mixing ratio experiments, N4–6. The horizontal axis is the time since the beginning of oxidation. For all except experiment L1, the light strength was identical.

## 7  Conclusions

The secondary organic aerosol yields of benzyl alcohol determined in this study range from 0.35 to 0.99. McDonald et al. (2018), who found that volatile chemical products might contribute very significantly to SOA formation in cities like Los Angeles, estimated a SOA yield of $0.090 \pm 0.023$ for benzyl alcohol. Even in its upper limit, this is less than a third of the SOA yields found in this study. While benzyl alcohol is one of a number of compounds considered, the fact that the experimental results disagree significantly with the estimates made in accounting studies indicates that we could still be vastly underestimating or poorly predicting SOA yields from oxygenated species.

The benzyl alcohol mixing ratios used in this study (>130 ppb) exceed substantially those in the atmosphere. Especially since we have suggested that, at least initially, SOA growth may proceed in a mass-transfer-limited regime, this could be a problem for extrapolating these results to the behavior of benzyl alcohol in the atmosphere. However, the long reaction time and the flattening out of the SOA yields (Fig. 5) suggests that the SOA yield has reached equilibrium and would be the same regardless of the precursor concentration. Furthermore, Figs. 2, 8, and 9 all show the mass of benzyl alcohol reacted at the end of an experiment as a function of SOA yield and the relevant other variable (initial seed surface area concentration, temperatures,





constant NO mixing ratio, respectively). In none of these figures does the amount of benzyl alcohol correlate to observed SOA yield.

This is seen more clearly in Fig. 14, where panel a shows the set of experiments carried out under approximately the same initial conditions and panel b shows all the experiments with a calculated SOA yield given in Table 1. Even for the reproduction experiments (panel a), there are some differences in initial benzyl alcohol mixing ratios. But, these differences do not lead to a discernible trend in the observed SOA yield (in panel a nor panel b); if anything, there appears to be an increase in SOA yield as the initial benzyl alcohol ratio decreases and, if this trend were applied to extrapolation to the atmosphere, we would only

expect to see larger SOA yields in the atmosphere than those reported here.

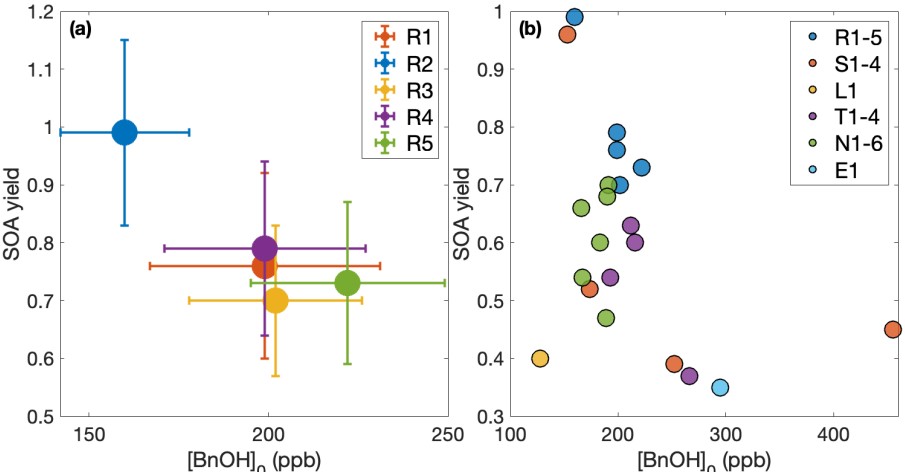

**Figure 14.** Effect of benzyl alcohol concentration on SOA yield. (a) The reproduction experiments (R1–5), which are all run under approximately the same conditions, with uncertainties. (b) All the experiments where a quantitative SOA yield is calculated. In both panels, we assume that $\omega = 0$. No trend is discernible in either panel.





As the SOA formed from benzyl alcohol has a NO mixing ratio dependence, a temperature dependence, and exhibits vapor-wall-deposition effects, it seems likely that other oxygenated compounds emitted from volatile chemical products will have similar behavior.

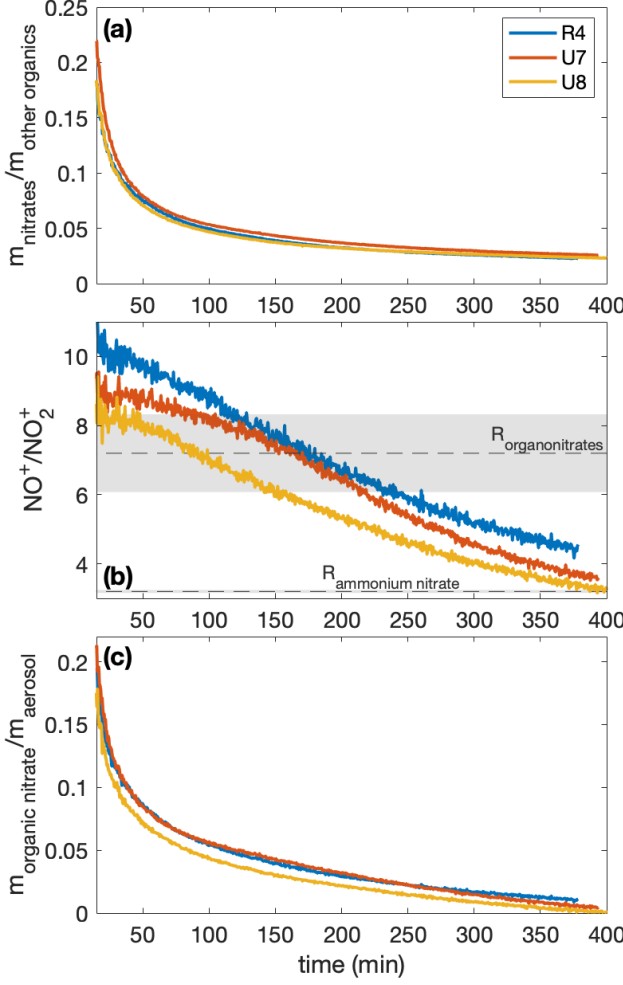

**Figure A1.** The mass ratios of (a) the nitrates to organics without nitrogen, (b) the $NO^+$ to the $NO_2^+$ signal from the AMS, and (c) the organonitrate to total organic aerosol mass for experiments R4, U7, and U8. All experiments were performed under similar initial conditions (291 K, $[NO]_0$ =71–77 ppb). Since the ratios are relevant only when there is a sufficient amount of aerosol present, the first 15 min after oxidation are not shown. In panel (b), the assumed organonitrate and ammonium nitrate NO to $NO_2$ ratios are shown as dashed lines with the uncertainty as the corresponding shaded region.





**Table A1.** Peak assignment for UPLC/ESI-Q-ToFMS analysis

| Retention Time (RT) | Mass | Error (mDa) | Molecular Formula | Compound |
|---|---|---|---|---|
| 3.484, 5.384 | 138.0147 | -3.9, -4.4 | $C_6H_5NO_3$ | |
| 3.857 | 137.0195 | -4.4 | $C_7H_6O_3$ | |
| 3.956, 4.485, 4.653 | 170.0047/2/5 | -4.2, -4.7, -4.4 | $C_6H_5NO_5$ | |
| 4.165, 4.180 | 184.0199/7 | -4.7/-5.0 | $C_7H_7NO_5$ | |
| 4.279 | 148.0352 | – | unassigned | |
| 4.348 | 121.0245 | -4.5 | $C_7H_6O_2$ | |
| 4.561 | 168.0250 | -4.7 | $C_7H_7NO_4$ | |
| 4.759 | 154.0096 | -4.4 | $C_6H_5NO_4$ | |
| 4.820, 5.079, 5.346 | 182.0047 | -3.9 | $C_7H_5NO_5$ | |
| 5.673 | 166.0097 | -4.3 | $C_7H_5NO_4$ | |
| 5.719 | 198.9991 | -4.2 | $C_6H_4N_2O_6$ | |

*Data availability.* Chamber data available upon request and through the Index of Chamber Atmospheric Research in the United States
(ICARUS).

*Author contributions.* JHS supervised the work. RSB did the filter collection, the UPLC-MS analysis, and conducted experiments U1 and U3–5. SMC designed the experiments, carried out the modeling, and did the rest of the data collection and analysis. SMC wrote the manuscript with contributions from RSB. All authors reviewed and edited the manuscript.



*Competing interests.* The authors declare that they have no conflict of interest.

*Acknowledgements.* The authors would like to thank Yuanlong Huang for his help with the SMPS and CIMS and for his general insight; Benjamin Schulze for his assistance with the AMS; Christopher Kenseth for his assistance with the AMS and UPLC; Lu Xu for his guidance on the AMS analysis; Nathan Dalleska for his help trouble-shooting chromatography methods and with UPLC analysis; John Crounse for his general help and for synthesis of $CF_3O^-$ for the CIMS; Paul Wennberg for the use of his FT-IR and for his insight into the chemistry of the system; Chris Cappa for very helpful comments on an early draft of this paper; and David Cocker III, Weihan Peng, and Qi Li for the use

of their SMPS for comparison purposes, suggestions for experimental conditions, and troubleshooting assistance. The project was funded by the California Air Resources Board (Contract #18RD009). SMC and RSB were funded by the National Science Foundation Graduate Research Fellowship program (#1745301).





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
