# Peer review of "Secondary Organic Aerosol Yields from the Oxidation of Benzyl Alcohol"

_Atmospheric Chemistry and Physics, 2020_

## Referee Comment (RC1) · Anonymous Referee #1 · 16 Jul 2020

This manuscript describes SOA formation from benzyl alcohol. Benzyl alcohol is a volatile chemical product (VCP). VCPs are a class of compounds that may be an important, but previously overlooked, source of SOA. The paper is timely because better laboratory data on SOA formation from VCPs is needed in order to constrain VCP SOA formation in chemical transport models.

Overall this paper is a comprehensive look at SOA formation from benzyl alcohol. Most of my comments focus on organization and structure. The paper seemed to jump back and forth between SOA yields (Section 3), SOA composition (Section 4), and SOA yields again (Section 5). I think it would be easier to follow if all of the discussion of yield was consolidated (e.g., switch sections 4 and 5). Additionally, I am not sure how much value is added by Section 6.

[Figure]

Line 88-89. I am not sure what this means "Multiple FT-IR samples were taken until each spectrum gave the same concentration; this was to ensure a minimal effect from any compound deposited on the instrument walls."

Line 103 - The TSI 3010 used t-butanol? I thought it used 1-butanol.

Figure 1a - why is SOA yield not shown for the first ∼10-15 minutes of the experiment?

Section 3.2.1 - were these wall loss experiments conducted as part of this study? Or are the authors recapping the experiments conducted in Charan et al 2018 and 2019?

Section 3.2.1 describes a fit for k_e and the electric field strength E. Some context would be helpful. For example, is E ∼ 2 V/cm "large" compared to the cases where E is effectively zero? Or is the E fit for each individual experiment effectively "small"?

To clarify - for Table 2 the "no correction" yield is the case where Delta_SOA is the suspended OA measured at the experiment end minus the initial OA mass (which is presumably ∼0)? I'm not sure this Table needs to be in the main text.

Figure 3 and 4 need more discussion. Are there systematic changes in OA composition at different T or [NO]? For example, in 3(c) I can see that the O:C ratios vary from experiment to experiment, but right now the reader needs to scrutinize the figure to see if there is any sort of trend between O:C and T.

Section 5 seems out of order. Perhaps put it with or immediately after section 3. Section 3 has a lot of detail on how SOA yield is calculated, and then Section 4 discusses SOA composition. It felt like a big jump and that I, as a reader, didn't have a sense that the SOA yields are high.

Figure 5 and 6 - Please specify what is meant by "reproduction experiments." I assume that these are nominally identical experiments, but it should be clarified.

Can you explain the choice of experiments shown in Figure 7? There are a number of experiments shown in Fig 6 that are not shown in Fig 7.

I am not sure what value Section 6 adds to the paper, especially since the model seems to perform poorly for many experiments. What do we learn from the modeling that cannot be learned directly from the data?

Regarding the mechanism discussion: I think one of the conclusions of section 6 is that SOA is not formed via the benzaldehyde channel. Does this mean that there is rough closure between SOA and gas-phase measurements? E.g., about a quarter of benzyl alcohol goes to benzaldehyde (no SOA formed). The rest forms products that can make SOA, and the SOA yield is in the neighborhood of 75%. Obviously we need to count the oxygen mass added, but to first order this adds up.

---

## Referee Comment (RC2) · Anonymous Referee #2 · 29 Jul 2020

Review of "Secondary Organic Aerosol Yields from the Oxidation of Benzyl Alcohol" by Charan et al.

Synopsis.

The researchers embark on an investigation of volatile consumer products (VCPs) and for the initial compound they consider the photooxidation of benzyl alcohol (BnOH). The experiments are run with NOx present in the system and at levels there are using virtually all the peroxy radicals formed react with NO to form alkoxy radicals. While the experiments are suitable for examining ozone formation, the authors have decided to focus on SOA formation. A chemical ionization mass spectrometer (CIMS) is adopted for measuring gas-phase oxidation products. For particle measurements, particularly for determining aerosol yields (Y), a scanning mobility particle analyzer (SMPS) has

been used to measure particle volumes which can then be converted to particle mass using a density (1.4). The results indicated extremely high yields for BnOH ranging from 0.35 – 1.0. The authors then examine values for the yield for as a function of several parameters important in atmospheric scenarios, such as NOx levels, temperature, amount of BnOH reacted, and seed surface area.

An extensive discussion is given regarding the effect of walls on the deposition of condensable organic compounds to particles lost to the chamber walls to provide a corrected yield, $\omega$ (gk: omega). This turns out to be a negligible correction compared to the uncertainty of the SOA yield determination. Following this additional wall corrections are enumerated and (presumably) justified. The most important is the actual particle loss to the chamber. This second correction is very important to the interpretation of the work because it leads to the authors selection of the ammonium sulfate (AS) aerosol seed surface area (and by extension mass) that the authors select for their experiments. They also consider gas-phase product loss to the walls but ultimately decide that it is unimportant. When all is said and done the extent of the corrections to particle loss is 10 – 20%. Ultimately, the authors conclude that an aerosol yield as a function of time converges on a single value for the parameterization considered and becomes independent on the amount of BnOH reacted.

The paper ends up by considering gas-phase mechanism relevant to the degradation of BnOH and possibly to aid in interpreting these very high yields, but sadly as the authors note (line 391), the considerations in Section 6 do not affect the SOA yields. However, the section does give some clue as to the plausibility of the reported results.

General comments.

This paper caught my eye from the SOA yields approaching unity under some conditions. The results certainly merit publication and the authors have generated a fairly comprehensive dataset for yields with NOx present under a reasonably wide set of conditions. However, I do believe that the paper needs considerable work to entice

physical scientists to read it and appreciate the significance of the results. My main comments will address organization, emphasis, assumptions, and consistency, in no particular order.

(1) I believe that much of the message and findings of the paper are lost in the inordinate focus on corrections to the calculated yield in Section 3. For example, we have a reasonably detailed discussion of the factor $\omega$ (loss of condensable organic products to particles on the chamber walls) only to find out on line 200 that this factor is basically irrelevant to the yield determination. For me, this comment suggests that this section is essentially appendix material. All of Section 3 really needs to be reduced to one or two pages. The only section that should be discussed in any detail is Section 3.2.1. Otherwise, just give the major findings from the section.

(2) The justification for use AS seed aerosol with very high surface areas is to compete for condensable organics with losses to the walls. This leads to a seed aerosol concentrations having masses probably $10 - 100$ times that found in the atmosphere. These conditions limit the relevance of these experiments to atmospheric conditions. However, I am more worried about the mechanism for SOA formation at high surface areas. In the atmosphere, adsorption of organics while present cannot compete with absorption of condensable organics into the organic mass already present. I wonder if this is the case in the present experiments. At these high surface areas, can the major process for SOA condensation be adsorption and not absorption. I think this is a subject that should be discussed in the paper or at least explicitly discounted by performing the necessary calculations.

(3) Are these results consistent with the partitioning approach developed at Caltech in the mid-1990's. Can these results be expressed in a two-product model described by Odum et al. (1996, 1997) with appropriate updates from Ng (2007) and possibly others?

(4) The error analysis needs to be addressed in more detail. Starting with the first fig-

ure, the most striking thing in the figure is the magnitude of uncertainty associated with the yield for this experiment. If this is representative, and presumably it is, it is likely due to random errors rather than systematic errors, since they are already discussed extensively in Section 3. Thus, it appears to me that the random error completely swamps out the systematic error. I am not sure how the authors expect to convince a modeler of the accuracy with this level of uncertainty. Is it possible that these considerable random errors are due to a relatively small SOA mass condensing onto seed aerosol of considerably higher mass leading to errors associated with the subtraction of two large numbers? The issue of random errors needs to be better addressed in the text.

(5) Section 4 seems to be an appendage to the paper. It adds little to the interpretations in the paper, is not mentioned in the Abstract or Conclusions, and for me distracts from the main message of the paper. Unless these results can be better incorporated into the yield consideration or perhaps in the mechanism discussion of Section 6, I would remove it from this paper, and perhaps base a separate paper on this data. As an alternative, can the data in Section 4 be used in conjunction with the mechanistic discussion of Section 6, in which case I would place it immediately before Section 6.

(6) I find much of Section 6 to be of little value the way it is currently presented. As noted earlier, the authors state that the actual composition of products is decoupled from the yield measurement. Thus, this section is more of academic interest than anything else. The photooxidation of BnOH in the presence of NOx simply follows analogous with mechanisms for photooxidation of toluene in terms of abstraction from the substituent group and addition to the aromatic ring.

(7) A mass balance estimation of measured and likely products from the NOx photooxidation of BnOH make it implausible that yields approaching unity are realistic. The mechanism of BnOH oxidation with NOx is reported to give benzaldehyde as a major product with a yield of 0.25 (Harrison and Wells, 2009; Bernard et al., 2013; see author refs). And yet the authors states on line 463 that benzaldehyde does not form condensable products. Benzaldehyde together with small ring-fragmentation products

also unlikely to give condensable products probably make up at least 50% of the initial reacted BnOH mass. With half of the mass unavailable for SOA formation, it is hard for me to justify SOA yields of 1 and greater. Moreover, Figure 14 suggests that an extrapolation to atmospheric BnOH levels would make the effective yield substantially greater than 1. Where is all the SOA mass coming from? It seems to me that this is a serious issue that the authors need to address.

(8) Two important parameters not tested are wet AS (metastable AS along the deliquescence curve; important for summertime SOA formation) and SOA yields in the absence of NOx. In addition, limitations for modeling the reported yields might be mentioned in the discussion or conclusions.

Some detailed comments by line number or other identifier.

Line 43 – Please comment on this sentence in the conclusions. Are the authors using the word "result" to mean BnOH SOA yield?

Line 65 – Since H2O2 absorbs to a negligible degree at 350 nm, it would be useful to know the value for the radiation output at FWHM of the blacklight source. It seems to me that the photolysis rate for H2O2 is of as much, if not more, value than that of NO2 given the initial conditions.

Line 96 – How about a consideration of systematic errors for the BnOH measurements? Also, if I understand this sentence correctly, Table 1 gives the variance of the reacted BnOH together with the initial BnOH. Why not simply have an additional column with the value for the reacted BnOH together with its variance, or is the claim that the variance in the reacted BnOH associated solely with its initial concentration?

Line 120 – This sentence comes out-of-the-blue? Can a reference be added?

Table 1 – A column for the initial NO2 concentration is desperately needed. Delete the final column if room is needed. (See below) The double-dagger appears to apply to all data in that column; is that correct? For column 7, what is the origin of the value for

[OH]?

Lines 176-198 – Most of this material should be relegated to an appendix which is already being used in this paper or a supplementary information section. The correction is of little use as noted.

Figure 1. Is the uncertainty shown in the grey for Panel (a) representative of all experiments? If so, a more critical discussion of this is needed. What experiment in Table 1 is represented in this figure?

Line 242 – Why is there a need to make an assumption? Is not the aerosol volume being measured during the background measurements?

Section 3.2.2 – This looks like another section for an appendix or SI, since this correction is not used in any fashion as noted in lines 273-275.

Table 2. Are the uncertainties given consistent with the random error shown in Figure 1a?

Section 4. I would move this section to that after Section 5 and try to tie this data to the discussion of the chemical mechanism.

Line 289 – Given that mass-transfer-limited is mentioned several times in the text, it would be valuable for the authors to give their explanation of the term. Is this term equivalent to saying that SOA formation is kinetically controlled?

Line 294 – What sort of particle-phase reactions do the authors have in mind? Oligomerization? Figure 3b. Why is the noise in this panel so much greater than in the other two? Figures 3 and 4 add little to the discussion of the reported yields and might be considered for elimination.

Figure 6. This figure is meant to be associated with Figure 5 but does a poor job doing so. I would simply give the [BnOH] at 200 minutes. And at what point does the system run out of NO? This may be the reason that more condensable products are

not produced after 100-200 minutes.

Figure 7. Could this figure be interpreted as showing that partitioning is occurring. Perhaps a figure of Y vs. M0 would be informative.

Figure 8. I see no evidence in the experimental section as to how the temperature of the chamber is controlled to allow this data to be obtained. Moreover, how is the temperature in the chamber held constant as the irradiation proceeds when radiative heating from the lights is continuously occurring? As I read the figure, a 20-degree increase in temperature reduces the yield from 1.0 to 0.4. Seems like a substantial effect.

Line 383; 393 – I would not call a description of the photooxidation of BnOH a theory. The "theoretical value" for benzaldehyde formation (29.6%) from Wang (2015) is based on assuming the branching ratio from the abstraction channel is 25% (Bernard et al., 2013) and then adding 4% from the OH-addition to the substituted position of the aromatic ring (the subject of the paper) determined by quantum chemistry calculations – hardly a theoretical value.

Figure 10. For the scheme presented here, I would explore the possibility of NO2 adding to the initial cyclohexadienyl radical to compete with O2 addition given in the third channel (0.41). I only mention this because nitro hydroxyaromatics could easily partition into the particle phase and could be amenable for detection by AMS.

Section 6.2. This model is largely an exercise in data fitting. A discussion of the important adjustable parameters and any physical significance would be appropriate. I'm not sure this section adds very much to the paper.

Line 423. Delete the word "precisely". I am not sure what the difference is between 'constrained' and 'precisely constrained'. In my view, the model is better characterized as unconstrained. My opinion of Section 6.2 is that it detracts from the main subject of the paper.

Figures 11 & 12 and Table 4. I would consider these as appendix-type materials even if Section 6.2 is to remain in the paper.

Line 463. Given the unconstrained nature of the aerosol model for the chamber, it does not surprise me that an effect from possible SOA products from benzaldehyde cannot be detected. This question (or assumption) regarding condensable products from benzaldehyde photooxidation is probably best addressed experimentally. Why not just do a photooxidation experiments using benzaldehyde as the reactant rather than BnOH? Note: I am not asking for additional experiment(s).

Figure 14. If the yield from this figure is extrapolated to ambient BnOH concentrations, a value of 2 or more would need to be used. Hardly seems plausible, especially since 25-50% of the BnOH products are non-condensable and given the yield McDonald uses of 0.09. Thus, these experiments would suggest that the yield should be increased by a factor of 20.

Line 503. Some text should be added to the appendix at least referring to Figures A1 and Table A1.

Line 546. Some indication of the availability of this report should be provided, if possible.

---

## Author Comment (AC1) · 20 Aug 2020

**Response to Reviews: Secondary Organic Aerosol Yields from the Oxidation of Benzyl Alcohol**

August 20, 2020

**Reviewer 1**

This manuscript describes SOA formation from benzyl alcohol. Benzyl alcohol is a volatile chemical product (VCP). VCPs are a class of compounds that may be an important, but previously overlooked, source of SOA. The paper is timely because better laboratory data on SOA formation from VCPs is needed in order to constrain VCP SOA formation in chemical transport models.

Overall this paper is a comprehensive look at SOA formation from benzyl alcohol. Most of my comments focus on organization and structure. The paper seemed to jump back and forth between SOA yields (Section 3), SOA composition (Section 4), and SOA yields again (Section 5). I think it would be easier to follow if all of the discussion of yield was consolidated (e.g., switch sections 4 and 5). Additionally, I am not sure how much value is added by Section 6.

Thank you for all your comments; each is addressed in blue text. We reorganized the paper by consolidating Sects. 4 and 5 so that the parts of Sect. 4 that were important

for discussing SOA yields now appear alongside the actual SOA yield discussion. We also removed some of Sect. 6.3 and moved the rest to what was Sect. 5 (and is now Sect. 4). The rest of what was Sect. 6 is now in the Appendix.

Specific Comments

Line 88-89. I am not sure what this means "Multiple FT-IR samples were taken until each spectrum gave the same concentration; this was to ensure a minimal effect from any compound deposited on the instrument walls."

The CIMS was calibrated using measurements from an FT-IR. However, to use the FT-IR requires preparing benzyl alcohol in a small Teflon bag and then sampling by flowing the sample from the small bag into the instrument enclosure. Since this involves flowing through Teflon lines and the sampling enclosure itself (either of which might adsorb or absorb benzyl alcohol), multiple FT-IR samples were taken in succession until all adsorption sites were filled and/or an equilibrium was reached within the sampling enclosure such that the gas-phase concentration in the sampling enclosure matched the gas-phase concentration in the small Teflon bag. To clarify this, the text was changed to: "Multiple FT-IR samples were taken in succession until there were consistent spectra; this was to ensure a minimal effect from any compound deposited on the FT-IR instrument walls or sampling lines."

Line 103 - The TSI 3010 used t-butanol? I thought it used 1-butanol.

Yes, it is 1-butanol, this was a typo. It has been changed to "butanol" in the manuscript.

Figure 1a - why is SOA yield not shown for the first ~10-15 minutes of the experiment?

For all the experiments that show SOA yield, we plot the SOA formed (calculated either with the assumption that $\omega = 0$ or $\omega = 1$) divided by the benzyl alcohol reacted. These two values are shown in Figure 1b. Since at time 0, when the lights are turned on, no

benzyl alcohol has reacted (and so the denominator of the SOA yield calculation is 0 or nearly 0), the measurement noise has a large effect on the SOA yield for the first few minutes. This can make it look like the SOA is incredibly large. To avoid this confusion, and to zoom in on the part of the curve that is relevant, we have removed the first few minutes from the SOA yield plot. We also do this (for the same reason) for what are now Figs. 3a, 4, and 7. Additionally, for Figs. 6, 10, and A1, which show data from the AMS, we similarly remove the first few minutes because the amount of organic aerosol present was insufficient to produce an adequate signal to see visualize these ratios.

In the caption for Fig. 1, we added the following sentence: "Due to low signal at the beginning, the first 10 min of the experiment are not shown." In the Fig. 3 caption, we add: "In panel (a), the first 30 min of oxidation are removed due to low signal and large uncertainties in SOA yield." In the caption of Fig. 4, we add "The first 30 min of oxidation are omitted due to low signal and large noise at the beginnings of the experiments." The captions of Fig. 6 (which was Fig. 3), Fig. 10 (which was Fig. 4), and Fig. A1 still have the sentence: "Since the ratios are relevant only when there is a sufficient amount of aerosol present, the first 15 min after oxidation are not shown."

Section 3.2.1 - were these wall loss experiments conducted as part of this study? Or are the authors recapping the experiments conducted in Charan et al 2018 and 2019?

All the particle wall loss experiments discussed in Section 3.2.1 were conducted for this study. For a further explanation: Caltech has two chambers: one is used for experiments without $NO_x$ and one is used for experiments with $NO_x$. The experiments reported and discussed in Charan et al. 2018 were all conducted in the $NO_x$-free chamber, which is 19.0 $m^3$. All the experiments in this paper, including these particle wall loss calculations, are conducted in the chamber with $NO_x$, the volume of which is 17.9 $m^3$. Both chambers are located in the same enclosure. Charan et al. 2019 did not report any new experiments, but references those conducted in Charan et al. 2018 (all in the 19 $m^3$ chamber) and experiments conducted in Schwantes et al. 2019.
To clarify this, the phrase "as outlined in Charan et al. (2018)" was changed to "following the protocol in Charan et al. (2018). The phrase "for this study" was added to the first and fourth paragraphs in Sect. 3.2.1.

Section 3.2.1 describes a fit for k_e and the electric field strength E. Some context would be helpful. For example, is E âĹij 2 V/cm "large" compared to the cases where E is effectively zero? Or is the E fit for each individual experiment effectively "small"?

Immediately after this description of field strength, context is now given: "Note that this is small: over 20 h of solely particle wall deposition and coagulation for an initial surface area concentration of $2.7 \times 10^3$ m$^2$ cm$^{-3}$ and a lognormal distribution centered around ∼125 nm, an $\bar{E} = 2.5$ V cm$^{-1}$ gave a number concentration 86% of that when $\bar{E} = 0$ (Charan et. al., 2018). A characteristic value for a chamber with charge is ∼45 V cm$^{-1}$ (McMurry and Rader, 1985). This chamber, unlike many with larger values of $\bar{E}$, is constantly suspended and does not touch the enclosure walls."

To clarify - for Table 2 the "no correction" yield is the case where Delta_SOA is the suspended OA measured at the experiment end minus the initial OA mass (which is presumably âĹij0)? I'm not sure this Table needs to be in the main text.

Yes, this is correct. This sentence is added to clarify: "Table 1 shows the SOA yield that would be calculated assuming that no particles were lost to the chamber walls during the experiment: this is simply the difference between the measured aerosol mass at the end of the experiment and that at the beginning, divided by the total reacted benzyl alcohol mass." Additionally, what was Table 2 is now merged with Table 1, since it is not sufficiently critical to merit its own table.

Figure 3 and 4 need more discussion. Are there systematic changes in OA composition at different T or [NO]? For example, in 3(c) I can see that the O:C ratios vary from experiment to experiment, but right now the reader needs to scrutinize the figure to see if there is any sort of trend between O:C and T.

[Figure]

Fig. 3 and 4 are now Fig. 6 and 10, respectively. In the Fig. 6 caption, we added "At higher temperatures, O:C ratios are larger and H:C ratios tend to be smaller. There is also a slight increase in the $NO_x^+$ ratio with temperature." In the Fig. 10 caption, we add: "The lower NO experiments have a higher O:C ratio later in the experiment than the high NO cases; no trend is obvious in H:C ratios."

We integrated a lot of what was Sect. 4 into Sect. 5 (see response to your next question) and consolidated the discussion of the aerosol from both sections into the text around the two figures. The following paragraph now appears (in Sect. 4.4), just before the introduction of Fig. 6: "A higher SOA yield at lower temperatures is also supported by Fig. 6, which shows how the chemical makeup of the aerosol is different for aerosol formed at different temperatures: the O:C ratio is higher and the H:C ratio is lower on aerosol formed at higher temperatures, meaning that more volatile compounds that might condense at lower temperatures (and have a smaller O:C ratio and a lower H:C ratio) do not condense at the higher temperature (panels a and c). Though the difference is slight, there is a trend for a larger $NO_x^+$ ratio (panel b) and, correspondingly, a larger mass fraction of organonitrates at higher temperatures. The former indicates that the organonitrates may be less volatile than other nitrogen-containing compounds that may condense into the aerosol phase (including, potentially, inorganic ammonium nitrate). The latter suggests that the gas-phase branching may be different. It may be that fewer organonitrates are formed at lower temperatures." Similarly, discussion of Fig. 10 is now located alongside the figure in Sect. 4.5.

Section 5 seems out of order. Perhaps put it with or immediately after section 3. Section 3 has a lot of detail on how SOA yield is calculated, and then Section 4 discusses SOA composition. It felt like a big jump and that I, as a reader, didn't have a sense that the SOA yields are high.

We want to discuss the aerosol chemical composition alongside the SOA yield, so we have now integrated some of the aerosol composition discussion with the aerosol yield (and present the SOA yield first to avoid jarring the reader). The discussion of the

change in aerosol chemical composition, which was the crux of Sect. 4, is now briefer and in Sect. 4.6 (which is at the end of what was previously Sect. 5).

Additionally, we removed some of what was Sect. 6.3 and moved the rest (including what was Fig. 13 and now is Fig. 8) to the discussion of SOA yield at different temperature so that we can discuss the changing gas-phase chemistry along with the changing aerosol-phase chemistry when analyzing the changes in SOA yield.

Figure 5 and 6 - Please specify what is meant by "reproduction experiments." I assume that these are nominally identical experiments, but it should be clarified.

In Fig. 5 (now Fig. 4), the description of panel (a) was changed to: "(a) experiments run under approximately identical conditions." Fig. 6 has been removed (see response to Reviewer 2's comments). In Appendix C3 (previously Sect. 6.2), the phrase is changed to "experiments run under approximately identical conditions except for initial seed surface area concentrations (S2–4 and R1–4)." The caption of Fig. C2 (previously Fig. 11) now has the phrase "(a) similar experiments" and to the fourth paragraph of Appendix C3 (previously Sect. 6.2), we add the phrase: "where the reproduction experiments are those performed under very similar initial conditions." The Fig. 8 (previously Fig. 13) caption is changed to "(a) similar experiments, R1–5." The sentence in the fourth paragraph of what is now Sect. 5 (was line 497) is changed to "Even for experiments R1–5, designed to be nearly identical." In Fig. 11 (previously 14), the descriptor "reproduction" is removed.

Can you explain the choice of experiments shown in Figure 7? There are a number of experiments shown in Fig 6 that are not shown in Fig 7.

We removed Fig. 7 in place of what is now Fig. 3. Figure 4a (like what was Fig. 7) only includes experiments run under approximately the same initial conditions (experiments R1–5). Experiment L1, which was conducted at $\sim 8\%$ of the light strength of the other experiments but otherwise has the same initial conditions as experiments R1–5 is included to demonstrate consistency at a lower fraction of the $[\text{BnOH}]_0$ reacted. In Fig.

3a, this is now clearer because it is represented in terms of organic aerosol ($\Delta SOA_{meas}$) present instead of in terms of the fraction of the initial benzyl alcohol reacted. Additionally, we want to demonstrate that for experiments R1–5 (not for L1, since it has not had enough time to react), the yield is asymptotically approached as $\Delta SOA_{meas}$ increases. This was supposed to be the point of Fig. 7, which shows this asymptote in terms of the fraction of reacted benzyl alcohol, but this is now clearer.

Since we do not expect to see the same SOA yields for the experiments run under different initial conditions (such as temperature for T1–4 and NO concentration for N1– 6), consistency across $\Delta SOA_{meas}$ (previously, fraction of reacted benzyl alcohol) and SOA yield is not expected. Nevertheless, we now provide this information in Fig. 3b so that they can be compared. However, plotting the curves for each experiment made this figure too messy and so only the endpoints are shown in Fig. 3b.

I am not sure what value Section 6 adds to the paper, especially since the model seems to perform poorly for many experiments. What do we learn from the modeling that cannot be learned directly from the data?

Section 6 suggests, importantly, values for the parameters $\alpha_p$ and $\omega$. The accommodation coefficient, $\alpha_p \approx 10^{-2}$, determines the regime in which an experiment occurs. For some experiments, this small values of $\alpha_p$ suggests that mass-transfer limitations may be important. While the fact that $\omega \approx 0$ is expected, it is often a parameter under debate. We added Appendix C1 to explain the importance of these parameters and the purpose of the chamber simulations.

As this in not important enough to include in the main text, but we still believe it adds to the discussion, we have moved the bulk of this section to Appendix C. For what was Sect. 6.3, we completely removed the discussion of SOA yield from benzaldehyde and moved the consideration of hydroxybenzyl alcohol (HOBnOH) to Sect. 4.4 (the temperature-dependence of the SOA yield). We believe that the HOBnOH discussion provides evidence of differing gas-phase chemistry at different temperatures and so

we kept it in the main text.

Regarding the mechanism discussion: I think one of the conclusions of section 6 is that SOA is not formed via the benzaldehyde channel. Does this mean that there is rough closure between SOA and gas-phase measurements? E.g., about a quarter of benzyl alcohol goes to benzaldehyde (no SOA formed). The rest forms products that can make SOA, and the SOA yield is in the neighborhood of 75%. Obviously we need to count the oxygen mass added, but to first order this adds up.

This is difficult to say, because benzaldehyde concentrations were not measured during these experiments. The fact that there is shift in the HOBnOH concentrations depending on experimental conditions (temperature or [NO]) does indicate that the differences in observed SOA yields originate from the gas-phase products (either being benzaldehyde or intermediates or aerosol-forming products).

We do get rough mass-closure between the measured SOA yields and the predicted gas-phase branching ratios, though. The following paragraphs were added to the conclusion (Sect. 5):

"The one-product absorptive partitioning model predicted a mass-based stoichiometric coefficient of $\alpha \approx 0.97$ for oxidation products that partitioned into the aerosol phase. If we assume that these oxidation products can be described by very oxygenated rings with a molecular weight of 188 g mol$^{-1}$, then this corresponds to a mole-based branching ratio of 0.56. This is not much more than the 0.41 calculated by Wang (2015) for the formation of very oxygenated rings from benzyl alcohol oxidation (see Fig. C1 and Appendix C2). While the SOA yields calculated here appear high, they are not far from the those predicted in the gas-phase for the least volatile oxidation products.

"A molecular weight of 188 g mol$^{-1}$ for benzyl alcohol oxidation products also appears to be reasonable: these products would have an Oxygen-to-Carbon atom ratio of 0.86 (see Table C1), which is close to the ratios we see in Figs. 6 and 10 of as much as 0.95 and 0.83, respectively."

**Reviewer 2**

The researchers embark on an investigation of volatile consumer products (VCPs) and for the initial compound they consider the photooxidation of benzyl alcohol (BnOH). The experiments are run with $NO_x$ present in the system and at levels there are using virtually all the peroxy radicals formed react with NO to form alkoxy radicals. While the experiments are suitable for examining ozone formation, the authors have decided to focus on SOA formation. A chemical ionization mass spectrometer (CIMS) is adopted for measuring gas-phase oxidation products. For particle measurements, particularly for determining aerosol yields (Y), a scanning mobility particle analyzer (SMPS) has been used to measure particle volumes which can then be converted to particle mass using a density (1.4). The results indicated extremely high yields for BnOH ranging from 0.35 – 1.0. The authors then examine values for the yield for as a function of several parameters important in atmospheric scenarios, such as $NO_x$ levels, temperature, amount of BnOH reacted, and seed surface area.

An extensive discussion is given regarding the effect of walls on the deposition of condensable organic compounds to particles lost to the chamber walls to provide a corrected yield, $\omega$ (gk: omega). This turns out to be a negligible correction compared to the uncertainty of the SOA yield determination. Following this additional wall corrections are enumerated and (presumably) justified. The most important is the actual particle loss to the chamber. This second correction is very important to the interpretation of the work because it leads to the authors selection of the ammonium sulfate (AS) aerosol seed surface area (and by extension mass) that the authors select for their experiments. They also consider gas-phase product loss to the walls but ultimately decide that it is unimportant. When all is said and done the extent of the corrections to particle loss is 10 – 20%. Ultimately, the authors conclude that an aerosol yield as a function of time converges on a single value for the parameterization considered and becomes independent on the amount of BnOH reacted.

The paper ends up by considering gas-phase mechanism relevant to the degradation of BnOH and possibly to aid in interpreting these very high yields, but sadly as the authors note (line 391), the considerations in Section 6 do not affect the SOA yields. However, the section does give some clue as to the plausibility of the reported results.

Thank you for all the comments. We have addressed each in blue text.

General Comments

This paper caught my eye from the SOA yields approaching unity under some conditions. The results certainly merit publication and the authors have generated a fairly comprehensive dataset for yields with NOx present under a reasonably wide set of conditions. However, I do believe that the paper needs considerable work to entice physical scientists to read it and appreciate the significance of the results. My main comments will address organization, emphasis, assumptions, and consistency, in no particular order.

(1) I believe that much of the message and findings of the paper are lost in the inordinate focus on corrections to the calculated yield in Section 3. For example, we have a reasonably detailed discussion of the factor $\omega$ (loss of condensable organic products to particles on the chamber walls) only to find out on line 200 that this factor is basically irrelevant to the yield determination. For me, this comment suggests that this section is essentially appendix material. All of Section 3 really needs to be reduced to one or two pages. The only section that should be discussed in any detail is Section 3.2.1. Otherwise, just give the major findings from the section.

To reduce some of the excessive length, we did move the description of how to calculate the SOA yield when $\omega = 1$ to Appendix B. However, the SOA yield calculations and corrections are very important to understanding the true yields. As this is the first part of a series of VCPs that we hope to study, we prefer to keep the focus on SOA

yield corrections. To make this clear that this is one of the goals of the paper, we added this paragraph to the end of the introduction: "While the experiments described here were performed under conditions that minimize corrections required to extrapolate SOA yields to the atmosphere, historically these corrections could be quite significant Zhang et al. (2014). As the first compound studied in a set of VCPs, we devote Sect. 3 to a detailed discussion of the SOA yield calculation including possible corrections. Understanding these corrections is critical to ensuring that the SOA yields calculated are reasonable."

(2) The justification for use AS seed aerosol with very high surface areas is to compete for condensable organics with losses to the walls. This leads to a seed aerosol concentrations having masses probably 10 – 100 times that found in the atmosphere. These conditions limit the relevance of these experiments to atmospheric conditions. However, I am more worried about the mechanism for SOA formation at high surface areas. In the atmosphere, adsorption of organics while present cannot compete with absorption of condensable organics into the organic mass already present. I wonder if this is the case in the present experiments. At these high surface areas, can the major process for SOA condensation be adsorption and not absorption. I think this is a subject that should be discussed in the paper or at least explicitly discounted by performing the necessary calculations.

Thank you for bringing up this point. We added Sect. 4.1 to address the effect of adsorption:

"The uptake and growth of aerosol can occur either through adsorption or absorption of oxidation products. Generally, we think of secondary organic aerosol growth as governed by absorption, though adsorption is also possible, especially at the large surface area concentrations used in this study to reduce the effect of vapor-wall deposition. To estimate the relative effects of these two processes, we use the gas-particle partitioning coefficient given by (Pankow, 1994, 1987):

$$K_p = \frac{1}{p_L^0} \left[ N_s A_{tsp} R T e^{\Delta Q/RT} + \frac{f_{om} R T}{MW_{om} \gamma} \right] \tag{1}$$

where the first term comes from adsorption and the second from absorption. The absorbent vapor pressure, $p_L^0$ is in units of atm. If we assume that the molecular weight of the organic material $MW_{om} = 188$ g mol$^{-1}$ = $1.88 \times 10^8$ g mol$^{-1}$, which is the molecular weight of the major low-volatility oxidation product of benzyl alcohol calculated by Wang (2015); the activity coefficient of a compound in the organic phase is $\gamma = 1$; and the temperature is $T = 291$ K (matching that in experiment R1), the absorptive term is $\sim (1.3 \times 10^{-10}) f_{om}$ m$^3$ atm g$^{-1}$, where $f_{om}$ is the mass fraction of absorbing organic in the aerosol phase.

"The specific surface area of the particulate matter, $A_{tsp}$, changes little throughout experiment R1. At the beginning of the experiment, when particles are the smallest, $A_{tsp} \approx 0.14$ cm$^2$ g$^{-1}$. Using Eq. 60 from Pankow (1987), the surface concentration of sorption sites on an adsorbing surface is $N_{s,om} \approx 4.5 \times 10^{-10}$ mol cm$^{-2}$ for the organic phase and $N_{s,amm\ sulf} \approx 6.7 \times 10^{-10}$ mol cm$^{-2}$ for ammonium sulfate. Note that the calculation for the organic phase uses $\rho_{om} = 1.4$ g cm$^{-3}$. To get an upper-bound estimate of adsorption, if we take $N_s = N_{s,amm\ sulf}$, the adsorptive term is $\sim (2.2 \times 10^{-12}) e^{1.7\Delta Q}$ m$^3$ atm g$^{-1}$, where $\Delta Q$ is the enthalpy difference between desorption from the particle surface and vaporization of the pure liquid and has units of kcal mol$^{-1}$.

"To determine the relative importance of adsorption and absorption, we need $\Delta Q$ and $f_{om}$. For liquid-like adsorption, $\Delta Q \approx 0$, but for SOA from polycyclic aromatic hydrocarbons rings and organochlorines, $\Delta Q \approx$ 2–4 kcal mol$^{-1}$ and $\Delta Q \approx$ 1–2 kcal mol$^{-1}$, respectively (Pankow, 1987; Yamasaki et al., 1982). For experiment R1, $f_{om}$ is 0.1 by 10 min and 0.5 by 1 h. At the end of the experiment, $f_{om} = 0.8$.

"Depending on the value of $\Delta Q$, the length into the experiment at which adsorption is insignificant changes. If $\Delta Q \approx 0$, adsorption will be responsible for $< 15\%$ partitioning

10 min into the experiment. If $\Delta Q \lesssim 0.9$ kcal mol$^{-1}$, adsorption will be responsible for $< 15\%$ partitioning 1 h into the experiment, and if $\Delta Q \lesssim 1.2$ kcal mol$^{-1}$, adsorption will be responsible for $< 15\%$ partitioning at the end of the experiment. Note that, since prior to the commencement of oxidation, no aerosol growth is observed, the seed aerosol neither adsorbs nor absorbs benzyl alcohol."

(3) Are these results consistent with the partitioning approach developed at Caltech in the mid-1990's. Can these results be expressed in a two-product model described by Odum et al. (1996, 1997) with appropriate updates from Ng (2007) and possibly others?

We added Sect. 4.2 to address this question, along with Fig. 3. Figure 4a has the one- and two-product model fits shown. Unlike Odum et al. (1997), which found two products are required (but any more is redundant), we find that a one-product model performs quite well. Section 4.2 is:

"If absorption dominates gas-particle partitioning, the SOA yield would depend on the amount of organic material in the aerosol phase ($\Delta$SOA$_{\text{meas}}$, which varies with $f_{om}$) if equilibrium growth occurs, as is shown in Fig. 3 (Pankow, 1994; Odum et al., 1996). Traditionally, this partitioning is given by

$$Y = \Delta\text{SOA}_{\text{meas}} \sum_{i=1}^{n} \left( \frac{\alpha_i K_{om,i}}{1 + K_{om,i}\Delta\text{SOA}_{\text{meas}}} \right) \qquad (2)$$

where a one-product model has $n = 1$ and a two-product model has $n = 2$ (Pankow, 1994; Odum et al., 1996; Ng et al., 2007). The stoichiometric fraction of product $i$ in mass units is $\alpha_i$. $K_{om,i}$ is the absorptive partitioning coefficient for the organic phase for species $i$, which is $\frac{K_{p,i}}{f_{om}}$ from Eq. 2 (Odum et al., 1996).

"The two-product model does not improve from the one-product model (dotted curve in Fig. 3a), but only creates a very large non-volatile compound ($K_{om} >> 1$) that is formed in very small quantities ($\alpha << 1$) and the other compound nearly matches the

compound found in the one-product optimization. The one-product optimization gives $\alpha = 0.97$ and $K_{om} = 0.009$ if all points are equally weighted. If we only include the end points, this gives $\alpha = 1.05$ and $K_{om} = 0.005$.

"At $\gtrsim 500$ g m$^{-3}$, the SOA yield flattens out. This indicates that, above this $\Delta SOA_{meas}$, the partitioning coefficients for the oxidative products are sufficiently large (that is, the products are sufficiently non-volatile), that Y approaches the $\alpha$, the gas-phase stoichiometric fraction in mass units for the oxidation products (Ng et al., 2007)."

(4) The error analysis needs to be addressed in more detail. Starting with the first figure, the most striking thing in the figure is the magnitude of uncertainty associated with the yield for this experiment. If this is representative, and presumably it is, it is likely due to random errors rather than systematic errors, since they are already discussed extensively in Section 3. Thus, it appears to me that the random error completely swamps out the systematic error. I am not sure how the authors expect to convince a modeler of the accuracy with this level of uncertainty. Is it possible that these considerable random errors are due to a relatively small SOA mass condensing onto seed aerosol of considerably higher mass leading to errors associated with the subtraction of two large numbers? The issue of random errors needs to be better addressed in the text.

What was Sect. 3.2.3 (Effect of corrections on measured SOA yield) is now Sect. 3.3 (Errors in measured SOA yields) and includes the additional information:

"For experiment R1, the assumed uncertainty that comes from particle-wall-deposition is $\sim 8\%$. This dominates, for $\Delta SOA_{meas}$, the random and counting error. The total uncertainty in $\Delta SOA_{meas}$ for experiment R1 is, including the uncertainty in the aerosol density, the wall-deposition, and the random error, $\sim 9\%$.

"Most of the reported uncertainty in the SOA yield comes not from the wall-deposition correction, but from the uncertainty in the benzyl alcohol concentration. For experiment R1, the random error in the benzyl alcohol signal, measured during the background collection period, was 15%. Combined with the uncertainty of the calibration (6%), this

was a 16% uncertainty. This same error was applied to the concentration of benzyl alcohol measured at the end of the experiment. Since $\Delta[\text{BnOH}]_{\text{meas}} = [\text{BnOH}]_0 - [\text{BnOH}]_{t=end}$, the uncertainty of $\Delta[\text{BnOH}]_{\text{meas}}$ is 19.5%.

"With the 9% and 19.5% uncertainties in $\Delta\text{SOA}_{\text{meas}}$ and $\Delta[\text{BnOH}]_{\text{meas}}$, respectively, we get a 21% uncertainty in the final calculated SOA yield. Most of this comes from the precursor concentration.

"Uncertainty from vapor-wall deposition is not included in the calculated error, but any vapor-wall deposition would only decrease the fraction of organic aerosol observed. That is, the true $\Delta\text{SOA}_{\text{meas}}$ would be larger than the calculated $\Delta\text{SOA}_{\text{meas}}$. If experiments were not run at a sufficiently large aerosol surface area concentration to neglect the loss of gas-phase products to the chamber walls, the true SOA yield will only be larger than what is reported here."

Error is also discussed in Sect. 3.2.1 (Particle wall deposition) and the third paragraph of Sect. 2.3 (Particle-phase measurements) and the second-to-last paragraph of Sect. 2.2 (Gas-phase measurements).

(5) Section 4 seems to be an appendage to the paper. It adds little to the interpretations in the paper, is not mentioned in the Abstract or Conclusions, and for me distracts from the main message of the paper. Unless these results can be better incorporated into the yield consideration or perhaps in the mechanism discussion of Section 6, I would remove it from this paper, and perhaps base a separate paper on this data. As an alternative, can the data in Section 4 be used in conjunction with the mechanistic discussion of Section 6, in which case I would place it immediately before Section 6.

Please see the response to Reviewer 1's eighth specific comment (p. C6–7).

(6) I find much of Section 6 to be of little value the way it is currently presented. As noted earlier, the authors state that the actual composition of products is decoupled from the yield measurement. Thus, this section is more of academic interest than anything else.

[Figure]

The photooxidation of BnOH in the presence of $NO_x$ simply follows analogous with mechanisms for photooxidation of toluene in terms of abstraction from the substituent group and addition to the aromatic ring.

Please see the response to Reviewer 1's eleventh specific comment (p. C8).

(7) A mass balance estimation of measured and likely products from the $NO_x$ photooxidation of BnOH make it implausible that yields approaching unity are realistic. The mechanism of BnOH oxidation with NOx is reported to give benzaldehyde as a major product with a yield of 0.25 (Harrison and Wells, 2009; Bernard et al., 2013; see author refs). And yet the authors states on line 463 that benzaldehyde does not form condensable products. Benzaldehyde together with small ring-fragmentation products also unlikely to give condensable products probably make up at least 50% of the initial reacted BnOH mass. With half of the mass unavailable for SOA formation, it is hard for me to justify SOA yields of 1 and greater. Moreover, Figure 14 suggests that an extrapolation to atmospheric BnOH levels would make the effective yield substantially greater than 1. Where is all the SOA mass coming from? It seems to me that this is a serious issue that the authors need to address.

To address the first part of this question, we added the following paragraphs to the text: "The one-product absorptive partitioning model predicted a mass-based stoichiometric coefficient of $\alpha \approx 0.97$ for oxidation products that partitioned into the aerosol phase. If we assume that these oxidation products can be described by very oxygenated rings with a molecular weight of 188 g mol$^{-1}$, then this corresponds to a mole-based branching ratio of 0.56. This exceeds modestly the value of 0.41 calculated by Wang (2015) for the formation of very oxygenated rings from benzyl alcohol oxidation (see Fig. C1 and Appendix C2). While the SOA yields calculated here appear high, they are not far from the those predicted in the gas-phase for the least volatile oxidation products.

"A molecular weight of 188 g mol$^{-1}$ for benzyl alcohol oxidation products also appears to be reasonable: these products would have an Oxygen-to-Carbon atom ratio of 0.86

(see Table C1), which is close to the ratios we see in Figs. 6 and 10 of as much as 0.95 and 0.83, respectively."

The goal of Fig. 14 (now Fig. 11) was not to indicate that extrapolating to the atmosphere would lead to higher yields. The conditions in panel b are mostly different, and so should not indicate this. While there does appear to be a vague trend in panel a, where all experiments are run under roughly similar conditions, this trend is slight and drowned out by the error bars. The point of Fig. 14 was simply to show that there is no obvious trend in the other direction. The text of the caption reads "No trend is discernible in either panel" and in the text "But, these differences do not lead to a discernible trend in the observed SOA yield (in panel a nor panel b)."

(8) Two important parameters not tested are wet AS (metastable AS along the deliquescence curve; important for summertime SOA formation) and SOA yields in the absence of $NO_x$. In addition, limitations for modeling the reported yields might be mentioned in the discussion or conclusions.

The following sentences were added in the conclusion: "When extrapolating SOA yields to the atmosphere, one should note that all these experiments were conducted at $< 9\%$ relative humidity, which is far below the deliquescence point. Additionally, all experiments were conducted in the presence of $NO_x$. Care should be taken when extrapolating these conditions to humid and low-$NO_x$ environments."

Specific Comments

Line 43 – Please comment on this sentence in the conclusions. Are the authors using the word "result" to mean BnOH SOA yield?

Yes, this was changed to "SOA yields."

Line 65 – Since H2O2 absorbs to a negligible degree at 350 nm, it would be useful to know the value for the radiation output at FWHM of the blacklight source. It seems to

me that the photolysis rate for H2O2 is of as much, if not more, value than that of NO2 given the initial conditions.

The $H_2O_2$ photolysis rate was added: "Ultraviolet broadband lights centered around 350 nm were used to photolyze $H_2O_2$ with a photolysis rate of $j_{H_2O_2} \approx 4.7 \times 10^{-6}$ s$^{-1}$. This was calculated using the measured variation in irradiance with wavelength and the $NO_2$ photolysis rate, $j_{NO_2} = 6.2(\pm0.1) \times 10^{-3}$ s$^{-1}$, which was found using a 0.29 L quartz tube and the procedure outlined in Zafonte et al. (1977)."

Line 96 – How about a consideration of systematic errors for the BnOH measurements? Also, if I understand this sentence correctly, Table 1 gives the variance of the reacted BnOH together with the initial BnOH. Why not simply have an additional column with the value for the reacted BnOH together with its variance, or is the claim that the variance in the reacted BnOH associated solely with its initial concentration?

We added the percentage of the initial benzyl alcohol that reacts by the end of the experiment to the second column of Table 1. The way that the variance in the reacted BnOH is calculated is based on the 1 h period before oxidation commences. This is also the way that the variance of the initial BnOH is calculated, which is why the percent error will be the same for the initial and final benzyl alcohol concentrations. The reported error in the benzyl alcohol concentration included the random error and the uncertainty in the calibration process (of 6%). The random error dominates the uncertainty. We added this paragraph to what is now Sect. 3.3: 'Most of the reported uncertainty in the SOA yield comes not from the wall-deposition correction, but from the uncertainty in the benzyl alcohol concentration. For experiment R1, the random error in the benzyl alcohol signal, measured during the background collection period, was 15%. Combined with the uncertainty of the calibration (6%), this was a 16% uncertainty. This same error was applied to the concentration of benzyl alcohol measured at the end of the experiment. Since $\Delta[\text{BnOH}]_{\text{meas}} = [\text{BnOH}]_0 - [\text{BnOH}]_{t=end}$, the uncertainty of $\Delta[\text{BnOH}]_{\text{meas}}$ is 19.5%."

Line 120 – This sentence comes out-of-the-blue? Can a reference be added?

This was an unpublished comparison. We changed the sentence to: "Uncertainty in the particle size was assumed not to exceed 2 nm, as is typical."

Table 1 – A column for the initial NO2 concentration is desperately needed. Delete the final column if room is needed. (See below) The double-dagger appears to apply to all data in that column; is that correct? For column 7, what is the origin of the value for [OH]?

Except for experiment E1, the initial $NO_2$ concentration was 0 for every experiment, which is why a column is not added. To make this clearer, the note on experiment E1 has this sentence added: "All other experiments began with no initial $NO_2$."

The double dagger applies to those in the initial seed surface area column that do not include a variance. This is only U1, U3, U4, U7, and U8. The description of the double dagger now has this additional sentence: "This applies to experiments U1, U3, U4, U7, and U8."

In column 7, the reported number is $k_{OH+BnOH}$[OH] taken from a first-order exponential fit of the benzyl alcohol concentration. Because we do not directly measure the OH concentration and $k_{OH+BnOH}$ should be the same for all the experiments, this column is more of a measure of [OH] than it is of $k_{OH+BnOH}$. The footnote is added: "The reported value is from a first-order exponential fit of the benzyl alcohol decay."

Lines 176-198 – Most of this material should be relegated to an appendix which is already being used in this paper or a supplementary information section. The correction is of little use as noted.

This is now Appendix B.

Figure 1. Is the uncertainty shown in the grey for Panel (a) representative of all experiments? If so, a more critical discussion of this is needed. What experiment in Table 1 is represented in this figure?

The uncertainty in BnOH concentration is discussed in the third paragraph of Sect. 2.2. The uncertainty of the SOA is a combination of the systematic uncertainty in particle wall loss and Poisson counting statistics. The details of this are in Sect. 2.3. The uncertainty reported in Table 1 matches that shown in Figure 1. To make this more complete and clearer, we added a few paragraphs to Sect. 3.3 (see response to General Comment 4).

Line 242 – Why is there a need to make an assumption? Is not the aerosol volume being measured during the background measurements?

Yes, the aerosol volume is measured during the background collection period. Because of particle-wall deposition, the volume concentration of aerosol steadily decreases during this time. Because particle-wall-deposition rates are size-dependent, this is not an assumption but a verification that the particle-wall-deposition rate is correct.

Section 3.2.2 – This looks like another section for an appendix or SI, since this correction is not used in any fashion as noted in lines 273-275.

While the correction is not used, it only is not used because the conditions of the experiment are such that it is not important. Since most people do not test for this, we think it is important to explicitly mention in the text.

Table 2. Are the uncertainties given consistent with the random error shown in Figure 1a?

Yes, they are. This Table has been combined with Table 1, per Reviewer 1's suggestion. In Table 1, the error on the SOA yield is calculated as described in Sect. 3.3 (and in response to General Comment 4).

Section 4. I would move this section to that after Section 5 and try to tie this data to the discussion of the chemical mechanism.

Please see the response to Reviewer 1's eighth specific comment (p. C6–7).

Line 289 – Given that mass-transfer-limited is mentioned several times in the text, it would be valuable for the authors to give their explanation of the term. Is this term equivalent to saying that SOA formation is kinetically controlled?

Yes, this is the same. We changed what was line 289 (now the second sentence of Sect. 4.6) to "kinetically controlled." We also changed the phrase in the fourth paragraph of the conclusion to "kinetically controlled (or mass-transfer-limited) regime...." In Appendix C4, we added the sentence: "This is equivalent to saying that the system is kinetically controlled."

Line 294 – What sort of particle-phase reactions do the authors have in mind? Oligomerization? Figure 3b. Why is the noise in this panel so much greater than in the other two? Figures 3 and 4 add little to the discussion of the reported yields and might be considered for elimination.

The way that the $NO_x^+$ ratio is calculated involves more error because the $NO_2^+$ signal is much smaller than the carbon atom signal. Since the $NO_x^+$ ratio has $NO_2^+$ as its denominator, there is more noise associated.

We moved Figs. 3 and 4 to the SOA yield discussion and focused, in their interpretations, on the differences in O:C and H:C ratios at different temperatures or NO concentrations. We think they are valuable to the SOA-formation discussions.

Oligimerization seems very possible. We have added the following sentence to Sect. 4.6: "There may also be particle-phase chemical reactions, such as oligimerization (Gao et al., 2004), that leads to the change in O:C ratio throughout the experiment."

Figure 6. This figure is meant to be associated with Figure 5 but does a poor job doing so. I would simply give the [BnOH] at 200 minutes. And at what point does the system run out of NO? This may be the reason that more condensable products are not produced after 100-200 minutes.

We now report the initial concentration, the fraction that has reacted by the end of the

experiment, the length of the experiment, and the first-order exponential fit to [BnOH] in Table 1 and omit what was Fig. 6. The point of this figure was to show that most of the benzyl alcohol had reacted by the end of the experiment and the fraction reacted (in parentheses in the second column of Table 1) captures this information.

In experiment R1, we observe NO close to 0 ppb around 200 min. Though, since we also see $NO_2$ and the lights are still on, the concentration is not 0 even at the end of the experiment. In experiments N1–6, where we constantly inject NO and it never runs out, we also see little condensable matter forming after that point. After 200 min, for experiment R1, 75% of the precursor has reacted and only 10% reacts for the rest of the experiment. This may be what is responsible for the SOA yield approaching a single value. It could also be that, after $\sim 200$ min, enough organic material has formed that the amount of organic material no longer matters. That is, in Eq. 3, as $\Delta SOA_{meas}$ increases, $Y \to \alpha$.

Figure 7. Could this figure be interpreted as showing that partitioning is occurring. Perhaps a figure of Y vs. M0 would be informative.

We added Fig. 3 and a discussion on absorptive partitioning (Sect. 4.2). What was Fig. 7 has now been removed, as the new Fig. 3 makes the point that Fig. 7 previously did.

Figure 8. I see no evidence in the experimental section as to how the temperature of the chamber is controlled to allow this data to be obtained. Moreover, how is the temperature in the chamber held constant as the irradiation proceeds when radiative heating from the lights is continuously occurring? As I read the figure, a 20-degree increase in temperature reduces the yield from 1.0 to 0.4. Seems like a substantial effect.

The chamber is located within a temperature-controlled enclosure. The enclosure is heated or air conditioned depending on the desired temperature and can be set from 15 to 40 ℃. The lights are located behind Teflon that is flushed with air, so that turning

them on minimally affects the chamber temperature (temperature rose by $<3$ °C due to the lights in the first 90 min of oxidation). The reported temperature is the temperature averaged over the duration of the experiment. The following phrase was added: "All experiments were performed in batch mode in the Caltech 17.9 m$^3$ FEP Teflon-walled Environmental Chamber, which hangs in a temperature-controlled enclosure."

Line 383; 393 – I would not call a description of the photooxidation of BnOH a theory. The "theoretical value" for benzaldehyde formation (29.6%) from Wang (2015) is based on assuming the branching ratio from the abstraction channel is 25% (Bernard et al., 2013) and then adding 4% from the OH-addition to the substituted position of the aromatic ring (the subject of the paper) determined by quantum chemistry calculations – hardly a theoretical value.

We chose "theory" to differentiate it from experimental, but have now changed the word that was on line 393 to "calculated" and the subsection title to "Gas-phase reactions." Note that this is now in Appendix C.

Figure 10. For the scheme presented here, I would explore the possibility of NO2 adding to the initial cyclohexadienyl radical to compete with O2 addition given in the third channel (0.41). I only mention this because nitro hydroxyaromatics could easily partition into the particle phase and could be amenable for detection by AMS.

This very well could be a mechanism for formation of nitroaromatics, which we do detect. We add some more discussion of nitroaromatics to the third-to-last paragraph of Sect. 4.5. We also add this sentence to Sect. 4.5: "Nitroaromatics could also form from the addition of NO$_2$ to a radical intermediate, as has been suggested as the formation mechanism for nitrocatechols from laboratory studies of *m*-cresol (Iinuma et al., 2010)."

Section 6.2. This model is largely an exercise in data fitting. A discussion of the important adjustable parameters and any physical significance would be appropriate. I'm not sure this section adds very much to the paper.

As this is a less critical part of the paper, it has been moved to Appendix C. Section C1 adds some context for the importance of the parameters: "To interpret the SOA yields and extrapolate them to the atmosphere, there are a few parameters that are useful. To understand the degree of kinetic vs. quasi-equilibrium growth, the accommodation coefficient to suspended particles, $\alpha_p$, is useful; as $\alpha_p$ approaches 1, the system becomes closer to quasi-equilibrium growth.

"While the difference in the assumed SOA yield between the case where gas-phase oxidation products produced in the chamber bulk readily partition onto particles deposited on the chamber wall ($\omega = 1$) and the case where the particles cease to participate in partitioning once deposited ($\omega = 0$) is slight, the general assumption is that $\omega = 0$ and any verification of that is useful for understanding chamber data. While we do not calculate $\omega$ here, if the accommodation coefficient to particles deposited on the chamber walls ($\alpha_{pw}$) is $\sim 0$, that indicates that $\omega \approx 0$."

Line 423. Delete the word "precisely". I am not sure what the difference is between 'constrained' and 'precisely constrained'. In my view, the model is better characterized as unconstrained. My opinion of Section 6.2 is that it detracts from the main subject of the paper.

This word (now in the third paragraph of Appendix C3) has been removed. The entire section is now in the Appendix.

Figures 11 12 and Table 4. I would consider these as appendix-type materials even if Section 6.2 is to remain in the paper.

These are now all in the Appendix and became Figs. C2 and C3 and Table C2.

Line 463. Given the unconstrained nature of the aerosol model for the chamber, it does not surprise me that an effect from possible SOA products from benzaldehyde cannot be detected. This question (or assumption) regarding condensable products from benzaldehyde photooxidation is probably best addressed experimentally. Why

not just do a photooxidation experiments using benzaldehyde as the reactant rather than BnOH? Note: I am not asking for additional experiment(s).

These experiments have been done and found little or no SOA formed from benzaldehyde. We have removed this section from the paper as it no longer relevant to the rest of the discussion here.

Figure 14. If the yield from this figure is extrapolated to ambient BnOH concentrations, a value of 2 or more would need to be used. Hardly seems plausible, especially since 25-50% of the BnOH products are non-condensable and given the yield McDonald uses of 0.09. Thus, these experiments would suggest that the yield should be increased by a factor of 20.

Please see the response to your General Comment 7.

Line 503. Some text should be added to the appendix at least referring to Figures A1 and Table A1.

Text that previously appeared in the text but that directly discusses Fig. A1 and Table A1 is now in Appendix A.

Line 546. Some indication of the availability of this report should be provided, if possible.

The url of this report was added to the citation. It now reads: "Carter, W. P. L., Malkina, I. L., Cocker III, D. R., and Song, C.: Environmental Chamber Studies Of VOC Species In Architectural Coatings And Mobile Source Emissions, Tech. rep., Center for Environmental Research and Technology, University of California, http://citeseerx.ist.psu.edu/viewdoc/summary?doi=10.1.1.81.305, 2005."

---

## Author Response (AR2)

**Response to Technical Corrections: Secondary Organic Aerosol Yields from the Oxidation of Benzyl Alcohol**

The responses to these comments and additions to the attached manuscript text are in blue. Figures 4, 6, and 8 were also changed in the attached manuscript.

Table 1 - experiments with the U prefix are different. They don't have the % of benzyl alcohol reacted or wall loss slopes. I didn't notice these experiments being called out in the Methods section as somehow different. Please clarify. 5

Experiments U1–8 all have some measurement errors that preclude calculating the SOA yield.

In Sect. 2.2, we added: "For the experiments labeled U3 and U7–8, there were errors with the CIMS measurements. Correspondingly, Table 1 does not report an initial benzyl alcohol concentration, a first-order exponential fit to the benzyl alcohol decay, or any SOA yields. These experiments are still included in Table 1 because their results are used to understand differences in chemical composition."

10

In Sect. 2.3, we added: "For experiments U1-8, there were issues with the particle-volume measurements or with the particlewall-deposition correction (see Sect. 3.2.1). While these experiments were used for the analysis of chemical composition, no SOA yields or wall-loss slopes are reported. Additionally, experiments U1, U3-4, and U7 report approximate initial seed surface area concentrations. There is no initial measured seed surface area concentration for experiment U8."

Figure 4 might be too small to see when typeset 15

We have changed this plot from 2x3 to 3x2. Since the width requirement stays the same, hopefully it is now more readable. We similarly changed Fig. 8.

Suggestion for Fig 6 - instead of each experiment being an independent color, you could show lines of progressive opacity of the same color from low to high temperature

When the opacity of this plot is changed, it is very difficult to see the individual curves. To make it easier to see the difference 20 in temperature, we have changed this plot to go from blue to red (as with Fig. 7).

Figure 9 caption notes that there were "constant" NO conditions. Does this mean that NO was continually added to the chamber during the experiment?

[revised manuscript text omitted]

(ω = 1)  |
|----------------------------------------|------------------------------------------------------------------|---------------------|---------------------------------------------------------------------------|----------------------------|---------------------------------------------------------------------------|------------------------------------------------------------------|-----------------------------------------------------------|----------------------------------------------------|-------------------|-------------------|
| R1/190321                              | 199±32 (82%)                                                     | 291.0±0.3           | 1.74±0.17                                                                 | 77.3±0.9                   | 0.048±0.050                                                               | $1.10 \pm 0.06$                                                  | 6.1 (86%)                                                 | 0.68 (89%)                                         | 0.76±0.16         | 0.79±0.16         |
| R2/190323                              | 160±18 (88%)                                                     | 290.9±0.3           | 1.98±0.18                                                                 | 77.4±0.8                   | -0.041±0.145                                                              | $1.03 \pm 0.06$                                                  | 6.5 (85%)                                                 | 0.87 (88%)                                         | 0.99±0.16         | 1.04±0.16         |
| R3/190312                              | 202±24 (95%)                                                     | 291.1±0.2           | 1.50±0.16                                                                 | 72.6±0.7                   | -0.027±0.042                                                              | $0.86 \pm 0.04$                                                  | 12.0 (73%)                                                | 0.54 (77%)                                         | 0.70±0.13         | 0.75±0.13         |
| R4/190319                              | 199±28 (85%)                                                     | 291.0±0.2           | $1.97 \pm 0.18$                                                           | 74.0±1.0                   | -0.009±0.076                                                              | $1.03\pm0.06$                                                    | 6.3 (85%)                                                 | 0.70 (88%)                                         | 0.79±0.15         | 0.83±0.15         |
| R5/190128                              | 222±27 (78%)                                                     | 291.2±0.2           | 2.19±0.21                                                                 | 93.7±0.7                   | -0.017±0.059                                                              | $0.71\pm0.03$                                                    | 8.8 (80%)                                                 | 0.58 (81%)                                         | 0.72±0.13         | 0.78±0.13         |
| S1/191219                              | 455±29 (60%)                                                     | 291.3±0.2           | $0.00 \pm 0.00$                                                           | 72.4±0.6                   |                                                                           | $0.49 \pm 0.03$                                                  | 5.3 (90%)                                                 | 0.41 (91%)                                         | $0.45 \pm 0.06$   | $0.47 \pm 0.06$   |
| S2/191002                              | 252±16 (85%)                                                     | 291.2±0.2           | 0.33±0.07                                                                 | ~96                        | -0.008±0.013                                                              | $0.99 \pm 0.04$                                                  | 6.3 (88%)                                                 | 0.34 (87%)                                         | $0.39 \pm 0.04$   | $0.41 \pm 0.04$   |
| S3/190930                              | 174±15 (83%)                                                     | $291.0 {\pm} 0.2$   | $0.64 \pm 0.10$                                                           | ~90                        | $0.016 \pm 0.017$                                                         | $1.17\pm0.05$                                                    | 4.5 (91%)                                                 | 0.48 (88%)                                         | $0.52{\pm}0.06$   | $0.54{\pm}0.06$   |
| S4/190325                              | 153±27 (82%)                                                     | $291.0 {\pm} 0.3$   | $5.47 \pm 0.32$                                                           | $77.8 \pm 0.8$             | $0.010 \pm 0.213$                                                         | $1.08\pm0.09$                                                    | 5.1 (88%)                                                 | 0.81 (84%)                                         | $0.96 {\pm} 0.25$ | $1.04{\pm}0.25$   |
| T1/190419                              | 216±30 (86%)                                                     | 296.7±0.4           | 2.33±0.21                                                                 | 75.6±0.9                   | $-0.069 \pm 0.062$                                                        | $1.44\pm0.07$                                                    | 5.0 (91%)                                                 | 0.54 (89%)                                         | $0.60 \pm 0.11$   | $0.63 \pm 0.11$   |
| T2/190417                              | 193±23 (89%)                                                     | 301.6±0.4           | 1.93±0.19                                                                 | 71.7±0.9                   | -0.012±0.060                                                              | $1.44\pm0.08$                                                    | 5.0 (91%)                                                 | 0.48 (88%)                                         | 0.54±0.09         | 0.57±0.09         |
| T3/190422                              | 212±34 (91%)                                                     | 306.6±0.4           | 2.76±0.23                                                                 | $76.9 \pm 0.7$             | $0.070 \pm 0.144$                                                         | $1.13\pm0.09$                                                    | 6.3 (89%)                                                 | 0.53 (84%)                                         | $0.63 \pm 0.13$   | $0.67 \pm 0.13$   |
| T4/190410                              | 266±43 (87%)                                                     | 311.6±0.5           | 2.12±0.2                                                                  | $80.4 \pm 0.8$             | $-0.013 \pm 0.114$                                                        | $1.18\pm0.08$                                                    | 5.5 (90%)                                                 | 0.32 (87%)                                         | $0.37 {\pm} 0.08$ | $0.39{\pm}0.08$   |
| N1/190408*                             | 191±27 (92%)                                                     | 291.1±0.3           | $2.00 \pm 0.19$                                                           | 4.8 (0.7-8)                | $0.056 \pm 0.101$                                                         | $1.27\pm0.05$                                                    | 5.0 (91%)                                                 | 0.63 (91%)                                         | $0.70 {\pm} 0.12$ | $0.73 {\pm} 0.12$ |
| N2/190403*                             | 190±35 (86%)                                                     | $290.9 \pm 0.3$     | $2.09 \pm 0.19$                                                           | 14.3 (8–18)                | $0.003 \pm 0.094$                                                         | $1.02\pm0.11$                                                    | 5.0 (88%)                                                 | 0.61 (90%)                                         | $0.68 {\pm} 0.16$ | $0.71 \pm 0.16$   |
| N3/190426*                             | 166±32 (79%)                                                     | $290.9 {\pm} 0.3$   | 2.71±0.23                                                                 | 64.0 (56-69)               | $0.027 \pm 0.070$                                                         | $0.77\pm0.06$                                                    | 6.0 (90%)                                                 | 0.56 (84%)                                         | $0.66 {\pm} 0.17$ | $0.70 {\pm} 0.17$ |
| N4/190401*                             | 183±17 (73%)                                                     | $291.0 \pm 0.3$     | $1.84{\pm}0.18$                                                           | 76.2 (52–106)              | $0.008 \pm 0.059$                                                         | $0.86 {\pm} 0.05$                                                | 5.0 (88%)                                                 | 0.54 (90%)                                         | $0.60 {\pm} 0.09$ | $0.63 {\pm} 0.09$ |
| N5/190424*                             | 167±19 (76%)                                                     | $290.9 \pm 0.3$     | $2.84{\pm}0.23$                                                           | 111.7 (103–118)            | $0.027 \pm 0.186$                                                         | $0.77\pm0.05$                                                    | 5.0 (91%)                                                 | 0.46 (85%)                                         | $0.54{\pm}0.10$   | $0.58{\pm}0.10$   |
| N6/190405*                             | 189±18 (76%)                                                     | $290.9 {\pm} 0.1$   | $1.78 \pm 0.18$                                                           | 200.6 (194-208)            | $0.000 \pm 0.082$                                                         | $0.76\pm0.03$                                                    | 5.0 (88%)                                                 | 0.42 (89%)                                         | $0.47 {\pm} 0.08$ | $0.50{\pm}0.08$   |
| $E1/200109^{\scriptscriptstyle \perp}$ | 295±18 (78%)                                                     | 291.1±0.2           | $2.83 \pm 0.22$                                                           | $1.4{\pm}1.0$              | $0.091 \pm 0.093$                                                         | $0.83 \pm 0.02$                                                  | 5.5 (89%)                                                 | 0.29 (82%)                                         | $0.35{\pm}0.05$   | $0.38{\pm}0.05$   |
| L1/190110                              | 135±12 (52%)                                                     | $285.78 {\pm} 0.03$ | $2.58 \pm 0.21$                                                           | $80.4{\pm}1.1$             | $0.033 \pm 0.009$                                                         | $0.115 \pm 0.002$                                                | 16.7 (58%)                                                | 0.10 (27%)                                         | $0.37{\pm}0.18$   | $0.51 {\pm} 0.18$ |
| U1/190327                              | 189±22                                                           | $290.9 {\pm} 0.2$   | ~4.03                                                                     | $81.1 {\pm} 0.7$           |                                                                           | $2.09 \pm 0.25$                                                  | 5.2 (88%)                                                 |                                                    |                   |                   |
| U2/190430                              | 136±20                                                           | 291.1±0.2           | $1.36 \pm 0.13$                                                           | $71.0 \pm 0.9$             |                                                                           | $1.16 {\pm} 0.07$                                                | 5.2 (91%)                                                 |                                                    |                   |                   |
| U3/190628                              |                                                                  | $291.2 \pm 0.4$     | ~1.48                                                                     | 77.7±0.9                   |                                                                           |                                                                  | 5.0 (91%)                                                 |                                                    |                   |                   |
| U4/190529                              | 139±26                                                           | 291.1±0.3           | ~5.40                                                                     | 70.7±0.7                   |                                                                           | $1.10{\pm}0.06$                                                  | 5.5 (90%)                                                 |                                                    |                   |                   |
| U5/190828                              | 325±20                                                           | $284.5 \pm 0.1$     | $1.70 \pm 0.14$                                                           | ~69                        |                                                                           | $0.19{\pm}0.01$                                                  | 5.4 (86%)                                                 |                                                    |                   |                   |
| U6/190428*                             | 152±25                                                           | 291.1±0.2           | 3.11±0.23                                                                 | 137.8 (133–144)            |                                                                           | $0.74{\pm}~0.06$                                                 | 5.9 (90%)                                                 |                                                    |                   |                   |
| U7/190225                              |                                                                  | $290.9{\pm}0.2$     | ~2.2                                                                      | 71.6±1.0                   |                                                                           |                                                                  | 6.6 (84%)                                                 |                                                    |                   |                   |
| U8/190227                              |                                                                  | $290.9 \pm 0.3$     |                                                                           | 76.9±0.9                   |                                                                           |                                                                  | 9.6 (78%)                                                 |                                                    |                   |                   |

[revised manuscript text omitted]